🔓 | **Open Peer Review** | Clinical Microbiology | Research Article

# Intra-hospital microbiome variability is driven by accessibility and clinical activities

Kaseba Chibwe,[1] Sathyavathi Sundararaju,[2] Lin Zhang,[1] Clement Tsui,[2,3,4,5] Patrick Tang,[2,3] Fangqiong Ling[1,6,7]

**ABSTRACT** The hospital environmental microbiome, which can affect patients' and healthcare workers' health, is highly variable and the drivers of this variability are not well understood. In this study, we collected 37 surface samples from the neonatal intensive care unit (NICU) in an inpatient hospital before and after the operation began. Additionally, healthcare workers collected 160 surface samples from five additional areas of the hospital. All samples were analyzed using 16S rRNA gene amplicon sequencing, and the samples collected by healthcare workers were cultured. The NICU samples exhibited similar alpha and beta diversities before and after opening, which indicated that the microbiome there was stable over time. Conversely, the diversities of samples taken after opening varied widely by area. Principal coordinate analysis (PCoA) showed the samples clustered into two distinct groups: high alpha diversity [the pediatric intensive care unit (PICU), pathology lab, and microbiology lab] and low alpha diversity [the NICU, pediatric surgery ward, and infection prevention and control (IPAC) office]. Least absolute shrinkage and selection operator (LASSO) classification models identified 156 informative amplicon sequence variants (ASVs) for predicting the sample's area of origin. The testing accuracy ranged from 86.37% to 100%, which outperformed linear and radial support vector machine (SVM) and random forest models. ASVs of genera that contain emerging pathogens were identified in these models. Culture experiments had identified viable species among the samples, including potential antibiotic-resistant bacteria. Though area type differences were not noted in the culture data, the prevalences and relative abundances of genera detected positively correlated with 16S sequencing data. This study brings to light the microbial community temporal and spatial variation within the hospital and the importance of pathogenic and commensal bacteria to understanding dispersal patterns for infection control.

**IMPORTANCE** We sampled surface samples from a newly built inpatient hospital in multiple areas, including areas accessed by only healthcare workers. Our analysis of the neonatal intensive care unit (NICU) showed that the microbiome was stable before and after the operation began, possibly due to access restrictions. Of the high-touch samples taken after opening, areas with high diversity had more potential external seeds (long-term patients and clinical samples), and areas with low diversity and had fewer (short-term or newborn patients). Classification models performed at high accuracy and identified biomarkers that could be used for more targeted surveillance and infection control. Though culturing data yielded viability and antibiotic-resistance information, it disproportionately detected the presence of genera relative to 16S data. This difference reinforces the utility of 16S sequencing in profiling hospital microbiomes. By examining the microbiome over time and in multiple areas, we identified potential drivers of the microbial variation within a hospital.

**KEYWORDS** hospital environment, microbiome, 16S RNA, blood agar culture, LASSO modeling, statistics, NICU, community-engaged science

Address correspondence to Patrick Tang, ptang@sidra.org, or Fangqiong Ling, fangqiong@wustl.edu.

Kaseba Chibwe and Sathyavathi Sundararaju contributed equally to this article. Author order was determined based on K.C. leading the initial draft writing.

Patrick Tang and Fangqiong Ling contributed equally to this article.

The authors declare no conflict of interest.

See the funding table on p. 16.

Each indoor built environment has a distinct microbiome, and the microbes that populate a building interact with the occupants in complex ways (1). People are often the source of the microorganisms, and microorganisms can conversely be transmitted from the environment to people (2). Among built environments, the microbiome of a hospital is especially important, because many patients in hospitals are particularly vulnerable to infection, and hospital-acquired infections (HAIs) can be fatal. In Europe alone, HAIs contribute to 37,000 deaths and €7 billion in associated costs annually (3). Hospitalized patients often have comorbidities or immunocompromising conditions that make them more susceptible to serious infections from not only pathogenic microorganisms but also opportunistic microorganisms, which are normally non-pathogenic to healthy individuals (4–6). Environmental microorganisms are one source of these preventable infections, and the rise in antibiotic resistance in bacteria further complicates the threat of HAIs. Antibiotic-resistant bacteria are difficult to treat and therefore contribute to higher mortality rates and healthcare costs (7).

Temporal and spatial dynamics in the hospital environment have been previously investigated using culture-dependent and culture-independent methods in different hospital settings globally (8–11). Microbes are actively transferred between the hospital-built environment and the occupants (12). Patients shed their bacteria into the environment when they stay in a room, and over time the surface microbiome changes dynamically in response (13). In contrast, newborns delivered via cesarean section are seeded with the hospital microbiome, and their initial microbiome can have lasting effects on their development (14). Moreover, special attention should be given to the bacteria in the neonatal intensive care unit (NICU). Newborns in that unit tend to be low-weight and high-risk and spend weeks to months in the hospital, so they are further subjected to the hospital's microbiome (15–17). Additionally, the effect of a newly built hospital opening on the microbiome has not been observed in the NICU, though it has been shown to increase bacterial abundance and change the community composition in other areas of the hospital (13).

The application of machine learning to 16S rRNA sequencing data can identify biomarkers through feature selection. Machine learning models account for inherent complexity and variation within the microbiome which would be otherwise overlooked by traditional statistical methods (18). The identified biomarkers can be potential pathogen-associated sensors, such as the association between *Rothia* sp. and severe acute respiratory syndrome coronavirus 2 (SARS-CoV-2) virus found in another hospital study (19). These types of cross-validated models can be integrated with infection control programs and should be further explored.

Although such studies have added depth to our collective knowledge of the hospital microbiome, most of them are limited to a specific area of the hospital such as the intensive care unit (10, 16, 17, 20–24). Studies that have extended to other areas of the hospital typically have focused on medical areas where patients are located, such as intensive care units, surgery wards, and emergency care units (8, 9, 13, 15, 19, 25–30). As a consequence, areas restricted to only healthcare workers are understudied. With regard to bacterial dispersal patterns, those restricted areas are as important as medical areas occupied by patients, because healthcare workers can act as vectors and transport microbes as they move throughout the day. Likely due to their higher mobility, Lax et al. found that the hand microbiomes of nursing staff in an inpatient hospital strongly reflected the hospital surface microbiome (13). This microbial exposure can also impact healthcare workers, whose gut microbiomes are known to be impacted by the hospital environment (31).

In this study, we aimed to investigate the bacterial microbiome of various surfaces in a newly built women's and children's hospital before and after patient occupancy in the NICU. In addition, we aimed to characterize the microbial community structure within different areas of the hospital by combining traditional culture-based bacteriology with next-generation sequencing of the 16S rRNA gene. Our main hypotheses were that (i) the bacterial microbial community could change before and after patient occupancy and

vary with hospital areas and (ii) specific microbes can be associated with specific hospital areas. To test the hypothesis on the occurrence of specific microbes within hospital areas, we created several machine-learning models to identify specific biomarkers within each hospital area. During the investigation, we engaged healthcare workers to assist in sampling their work environment to increase awareness of the significance of the hospital microbiome. The research presented here intends to widen the understanding of microbial dynamics in the hospital by expanding sampling into multiple areas restricted to healthcare workers and applying rigorous statistical and machine learning analyses. This new understanding can then in turn improve infection control practices in the hospital.

## MATERIALS AND METHODS

### Site description and experimental design

#### Site description

A hospital microbiome sampling campaign was conducted at Sidra Medicine, a 400-bed women's and children's hospital located in Doha, Qatar. The hospital serves over 250,000 patients every year. The hospital opened its outpatient clinic in May 2016. The inpatient facilities opened on 14 January 2018. The inpatient facilities consisted of nine levels and four towers (towers A, B, C, and D, Fig. 1A). Prior to January 2018, there were outpatient services in the inpatient facilities and also other activities in preparation for full operation.

#### Surface sampling from the NICU

Samples of surfaces were taken from different locations in the NICU at Sidra Medicine, before and after the hospital began inpatient operation. Samples were collected prior to full operation on 3 December 2017, and samples were collected during full operation on 13 March 2018, 3 months after the hospital opening (Fig. 1B).

#### Surface sampling from other hospital areas

During April and May of 2018, surface samples were taken in various areas defined by the hospital activity, including (i) pediatric intensive care unit (PICU), (ii) pediatric surgery

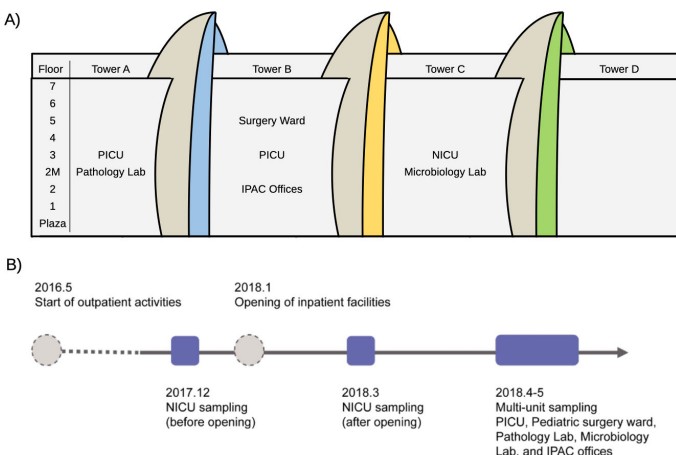

FIG 1 (A) Map of the levels and towers at Sidra Medicine. Areas that were sampled in this study are labeled. These include pathology lab (*n* = 20), infection prevention and control (IPAC) offices (*n* = 60), microbiology lab (*n* = 20), pediatric intensive care unit (PICU) (*n* = 20), NICU (before-opening: *n* = 16, after-opening: *n* = 25), and pediatric surgery ward (*n* = 36). (B) Sampling timeline and opening of outpatient and inpatient services. NICUs were sampled before and after the opening of inpatient services. All other areas were sampled after the opening of inpatient services.

ward, (iii) pathology lab, (iv) microbiology lab, and (v) infection prevention and control (IPAC) offices. Sampling was performed by 10 healthcare workers and volunteers working in the respective units. After an orientation on collection technique and definitions of high-touch surfaces, the volunteers sampled five surface sites of their choice weekly for 4 weeks (Fig. 1B).

## Sampling, amplicon sequencing, and culture-based analysis

### *Sampling and aliquoting*

Surface microbiome samples were collected using the ESwab collection kit (Copan Diagnostics, Italy). Samples were collected by trained healthcare workers following a written protocol. Surfaces were swabbed 10 times in three directions (horizontally, vertically, and diagonally) up to a maximum area of 30 cm × 30 cm. For smaller objects or surface areas, the entire available surface was swabbed. After a surface was sampled, specimens were sent immediately to the laboratory through the hospital pneumatic tube system. In the laboratory, samples were vortexed in the 400 L elution buffer from the ESwab collection kit and stored at −80°C until DNA extraction. For samples collected by healthcare workers, 100 L of the eluent was plated on sheep blood agars (MicroLab Medical, Qatar) prior to storage.

### *DNA extraction, 16S rRNA gene amplicon library preparation, and multiplexed sequencing*

Microbial DNA was extracted from the specimens using the NucliSENS easyMag system (bioMerieux, France). The sequencing libraries were prepared using the NEXTflex V1-V3 16S Amplicon-Seq Kit (Perkin-Elmer, MA) and purified using Agencourt Ampure XP magnetic beads (Beckman Coulter, CA). Purified amplicons were quantified using the Agilent TapeStation (Agilent, CA), pooled, and sequenced on the MiSeq equipment (Illumina, CA) to generate 2 × 300 bp paired-end reads. A negative water control was amplified and sequenced with all sequencing runs.

### *Bacterial culturing and matrix-assisted laser desorption/ionization time-of-flight*

A 100 L aliquot of each sample collected by healthcare workers was plated on sheep blood agar and incubated at 37°C for 18–24 h. Sheep blood agar was the cost-effective medium that would identify the most clinically relevant bacteria. The identification of bacteria followed standard clinical laboratory procedures where bacterial colonies were selected based on morphological differences and then identified using the Bruker Biotyper matrix-assisted laser desorption/ionization time-of-flight mass spectrometry (MALDI/TOF MS) system (Bruker, Germany) according to the manufacturer's protocol.

## Sequence denoising, amplicon sequence variant calling, taxonomic classification, and rarefaction

We performed quality filtering and denoising on the demultiplexed raw paired-end reads (2 × 300 bp) using the QIIME 2 platform (32). To achieve a quality score above 10, the forward reads were truncated at 285 bp; reverse reads were trimmed at 15 bp and truncated at 250 bp. The trimmed reads were denoised and merged using the DADA2 algorithm as implemented in the q2-dada2 plugin, which implemented a quality-aware model of Illumina amplicon errors and resolved differences as subtle as one nucleotide (33). Sequencing reads with a number of expected errors of more than two and detected chimeras were discarded. An amplicon sequence variant (ASV) table was thus generated. The denoising statistics generated from QIIME 2 are available in Table S1, and the read quality through a simulated pipeline is shown in Table S2 and Fig. S1. Taxonomy classification of the resulting ASVs was performed using a multinomial Naive Bayes classifier which was trained on a SILVA 138 reference database with the confidence

threshold set at 0.7 (default) (34). For completeness Reads that were not assigned to a domain (2.91%) or assigned to *Eukaryota* (0.12%), *Mitochondria* (0.74%), or *Chloroplast* (5.66%) were filtered out. The "isContaminant" function from the decontam R package was used to identify contaminant ASVs based on their frequency and the DNA concentration of the samples (35). The 29 contaminant ASVs were removed. All samples were rarefied to the same depth (1,973 reads).

## Metadata curation

The metadata included locations (towers, levels, and room numbers), area types (six areas as described earlier), room types, surfaces, and sampling dates. The sampling covered a wide range of surfaces as specified in Table S3 as "surface descriptions." We grouped these surface descriptions under five categories, which are door handles, keyboards, office electronics, floors, and screens. Surfaces that did not fit these five categories were labeled as "other surfaces" (Table S3).

## Diversity analysis

Alpha and beta diversity analyses were performed in R using the "phyloseq" package based on functions defined in "vegan" (36, 37). Alpha diversity was calculated as the observed richness, Shannon index, and Pielou's evenness using the "estimate_richness" function in phyloseq. In the analysis of NICU patient room surface microbiomes, the Wilcoxon rank-sum test was used to test if the average alpha diversity metrics in the surface microbiomes before and after opening were significantly different (implemented with the "wilcox.test" function in the R base functions). In the analysis of all other samples, the Kruskal–Wallis test (implemented with the "kruskal.test" function in the R base functions) was used to first test if the average alpha diversity metrics differed among the area types and surface categories. Upon detecting a significant difference among groups, a pairwise Wilcoxon rank-sum test was performed using the "pairwise.wilcox.test" function in the R base functions. The Benjamini-Hochberg procedure was implemented using the "p.adjust" function in R base functions to control false discovery.

Beta diversity was calculated with the "ordinate" function using Bray-Curtis dissimilarity on the log-transformed ASV relative abundances. Principal coordinate analysis (PCoA) was performed on the Bray-Curtis dissimilarity matrix. For completeness, non-metric multidimensional scaling was also performed on the Bray-Curtis dissimilarity (Fig. S7), PCoA was performed on the unweighted UniFrac, weighted Unifrac, and Jaccard dissimilarity matrices (Fig. S6). Permutational multivariate analysis of variance (PERMANOVA) was used to test if the centroids of the groups were significantly different (= 0.05). In the analysis of the NICU patient rooms, the groups were before and after opening samples. In the analysis of door handles, keyboard, and office electronic samples taken after opening, the groups were defined by area type, surface category, and number of days after opening in a single statistical test. When needed, pairwise PERMANOVA was performed as well, and the Benjamini-Hochberg method was applied to control false discovery.

## Machine learning models

Four machine learning models were utilized to distinguish area types from one another based on their sample ASV composition. The models used were Least Absolute Shrinkage and Selection Operator (LASSO) classification, support vector machines (SVM) with a linear kernel, SVM with a radial kernel, and random forest. The predictors were the log-transformed relative abundances of ASVs in the sample. For LASSO, the response variable was whether or not the sample originated from the area type being explored (binary class), and for SVM and random forest, the response variable was the area type from which the sample originated (multiclass). The data were split; 80% was used for training, and 20% was used for testing. Ten-fold cross-validation repeated 10 times was

implemented with the "train" function in the "caret" package on the training data (38). The hyperparameters were tuned by optimizing the accuracy. The performances of the resulting models were evaluated with the testing accuracy and Matthew's correlation coefficient (MCC). For the LASSO models, predictors with non-zero coefficients were identified as informative ASVs.

## RESULTS

### Citizen scientist engagement yielded broad coverage of the hospital

In this study, healthcare workers served as citizen scientists to collect hospital microbiome samples from high-touch surfaces of their choice. This novel approach allowed us to acquire samples across various areas, including the PICU, pediatric surgery ward, pathology lab, microbiology lab, and the office of the IPAC department. In addition, our research team sampled the NICU before and after the hospital opened. Diverse room types were covered, including patient rooms, nursing stations, and rooms in administrative areas (e.g., conference rooms, staff lounges, and locker rooms; Table S3A). A variety of surfaces were sampled, including door handles, keyboards, office electronics, and medical equipment (Table S3B). Door handles and keyboards were the most sampled (64 door handle samples and 37 keyboard samples prior to rarefaction; 59 door handle samples and 33 keyboard samples after rarefaction), suggesting these high-touch areas were commonly chosen sites for monitoring the hospital microbiome. Whereas most earlier studies have focused on one area, the broad coverage of multiple areas achieved in this study reflects the usefulness of engaging healthcare workers directly in a collective effort.

### Core microbiome analysis reveals microbes common across the hospital

After sequencing, quality filtering, and rarefying the microbe samples, a total of 178 samples were retained, generating 8,085 ASVs. We then identified common ASVs among the samples, which could be considered the core microbiome of the hospital, despite their spatial and temporal diversity. The relative abundances and prevalences of the ASV exhibited a positive correlation (*P*-value < 2.2e-16, Spearman = 0.672; Fig. S2 and S3); therefore, the core-satellite theory was applied (39). With a prevalence cutoff of 50%, the core microbiome consisted of 25 ASVs that represent 52.8% of the relative abundance.

The core community consisted of the following genera: *Cutibacterium*, *Chryseobacterium*, *Rhizobium*, *Corynebacterium*, *Pseudomonas*, *Micrococcus*, *Brevundimonas*, *Lawsonella*, *Empedobacter*, *Rothia*, *Comamonas*, *Enhydrobacter*, *Roseomonas*, *Peptoniphilus*, *Fusobacterium*, *Paracoccus*, and *Anaerococcus*. Among these organisms, *Cutibacterium* spp., *Corynebacterium* spp., *Micrococcus* spp., and *Anaerococcus* spp. are commonly associated with healthy human skin microbiota (40–43). Their high abundance highlights the importance of hospital environments in the dispersal of skin-associated microorganisms. *Pseudomonas* spp. are commonly found in natural and built environments, notably including pipelines for potable water (44). *Pseudomonas aeruginosa* strains are frequently reported as the etiologic agents of hospital-acquired infections including clusters in the NICU (7, 45, 46). *Chryseobacterium* spp. is found in diverse natural habitats and human microbiota. Though not directly detected in this study, certain strains of *C. indologenes* have been found in hospital-acquired infections and are increasingly recognized as an emerging pathogen affecting immunocompromised populations (47, 48), and *C. meningosepticum* is a known agent of neonatal meningitis and the most pathogenic species in the genus (49). *Fusobacterium* spp. are commonly found in the human oral, gastrointestinal, and genital microbiome. Though uncommon, *F. necrophorum* and *F. nucleatum* can cause Lemierre's syndrome, sepsis, and puerperal infections (50). Both *Paracoccus* spp. and *Brevundimonas* spp. are ubiquitous in the environment, however, the strains *P. yeei*, *B. diminuta*, and *B. vesicularis* are opportunistic pathogens and can cause nosocomial infections in patients (51, 52). Overall, the core microbiome is highly diverse, and the genera inhabiting the hospital surfaces potentially include pathogenic species.

**TABLE 1** Comparisons of means in alpha diversity metrics

| Grouping | Observed richness | | Shannon index | | Pielou's evenness | |
|---|---|---|---|---|---|---|
| | *P*-value | *P*-adjusted | *P*-value | *P*-adjusted | *P*-value | *P*-adjusted |
| (A) Wilcoxon rank sum tests on means of alpha diversity metrics in NICU patient rooms before and after opening | | | | | | |
| Before opening (*n* = 14) vs after opening (*n* = 14) | $1.77 \times 10^{-1}$ | $-^b$ | $3.26 \times 10^{-1}$ | $-$ | $5.42 \times 10^{-1}$ | $-$ |
| (B) Kruskal-Wallis tests on means of alpha diversity metrics in different surface categories and area types in all door handles, keyboard, and office electronic samples taken after the hospital opened for inpatient care (*n* = 120)$^a$ | | | | | | |
| Surface category | $1.10 \times 10^{-1}$ | $1.10 \times 10^{-1}$ | $2.86 \times 10^{-1}$ | $2.86 \times 10^{-1}$ | $5.74 \times 10^{-1}$ | $5.74 \times 10^{-1}$ |
| Area type | $\mathbf{9.84 \times 10^{-15}}$ | $\mathbf{1.97 \times 10^{-14}}$ | $\mathbf{8.57 \times 10^{-14}}$ | $\mathbf{1.71 \times 10^{-13}}$ | $\mathbf{1.67 \times 10^{-10}}$ | $\mathbf{3.35 \times 10^{-10}}$ |

$^a$Bonferroni correction was used to adjust *P*-values, and significant *P*-values are bolded (= 0.05).
$^b$–, not applicable.

## The hospital opening did not affect the NICU patient room microbiome

We examined a subset of 28 NICU patient room samples for temporal effects of the initiation of inpatient services on the surface microbiomes. Though there was an increase in alpha diversity after the hospital opened, no significant differences were detected in the number of observed ASVs (richness), the Shannon index (accounting for both richness and evenness), or Pielou's evenness (Wilcoxon tests *P*-value = 0.177, *P*-value = 0.326, *P*-value = 0.542; Table 1 ; Fig. S4). According to the PCoA of the Bray-Curtis dissimilarities, the transition to full operation did not significantly alter the community composition of the NICU patient rooms (PERMANOVA *P*-value = 0.723; Table 2; Fig. S5). These diversity metrics indicate that the hospital opening did not create a major change in the microbiome in the NICU patient rooms.

## The specific hospital area most strongly affects alpha and beta diversity relative to surface type and time

Recruiting healthcare workers allowed us to sample multiple hospital areas for 2 months. We subdivided and analyzed 120 samples of door handles, keyboards, and office electronics taken after the hospital opened. For each area (NICU, PICU, pediatric surgery ward, pathology lab, microbiology lab, and IPAC offices) and surface type (door handle, keyboard, and office electronics), we determined the alpha diversity of the bacterial communities by calculating the observed richness, the Shannon index, and Pielou's evenness. Across this data set, the average ASV richness was 153 [95% CI (146, 159)], the average Shannon index was 2.86 [95% CI (2.78, 2.94)], and the average Pielou's evenness was 0.578 [95% CI (0.566, 0.591)].

For all these alpha diversity metrics, the area types significantly differed from each other (Kruskal-Wallis tests *P*-value $< 10^{-10}$ for all metrics; Fig. 2; Table 1). Notably, the pediatric surgery ward, IPAC offices, and NICU exhibited the lowest diversities, and the

**TABLE 2** Comparisons of centroids in PCoA$^a$

| Grouping | Df | PERMANOVA | | | PERMDISP$^c$ | | |
|---|---|---|---|---|---|---|---|
| | | F value | R$^2$ | *P*-value | F value | *P*-value | *P*-adjusted |
| (A) PERMANOVA test on centroids and PERMDISP test on dispersions of PCoA in NICU patient rooms before and after opening | | | | | | | |
| Hospital opening | 1 | $7.23 \times 10^{-1}$ | $2.70 \times 10^{-2}$ | $7.32 \times 10^{-1}$ | $9.48 \times 10^{-2}$ | $7.69 \times 10^{-1}$ | $-^d$ |
| (B) PERMANOVA tests on centroids and PERMDISP test on dispersions of PCoA in different surface categories, area types, and days after opening in all door handles, keyboard, and office electronic samples taken after the hospital opened for inpatient care (*n* = 120)$^b$ | | | | | | | |
| Surface category | 2 | 2.42 | $2.89 \times 10^{-2}$ | $\mathbf{6.00 \times 10^{-3}}$ | $4.53 \times 10^{-2}$ | $9.45 \times 10^{-1}$ | 0.945 |
| Area type | 5 | $1.01 \times 10^{1}$ | $3.01 \times 10^{-1}$ | $\mathbf{1.00 \times 10^{-3}}$ | 1.50 | $2.07 \times 10^{-1}$ | 0.594 |
| Days after opening | 1 | $8.95 \times 10^{-1}$ | $5.36 \times 10^{-3}$ | $4.40 \times 10^{-1}$ | 1.05 | $3.96 \times 10^{-1}$ | 0.594 |

$^a$ASV relative abundances were square-root transformed prior to calculating the dissimilarity.
$^b$Bonferroni correction was used to adjust *P*-values, and significant *P*-values are bolded (= 0.05).
$^c$PERMDISP, permutational multivariate analysis of dispersion.
$^d$–, not applicable.

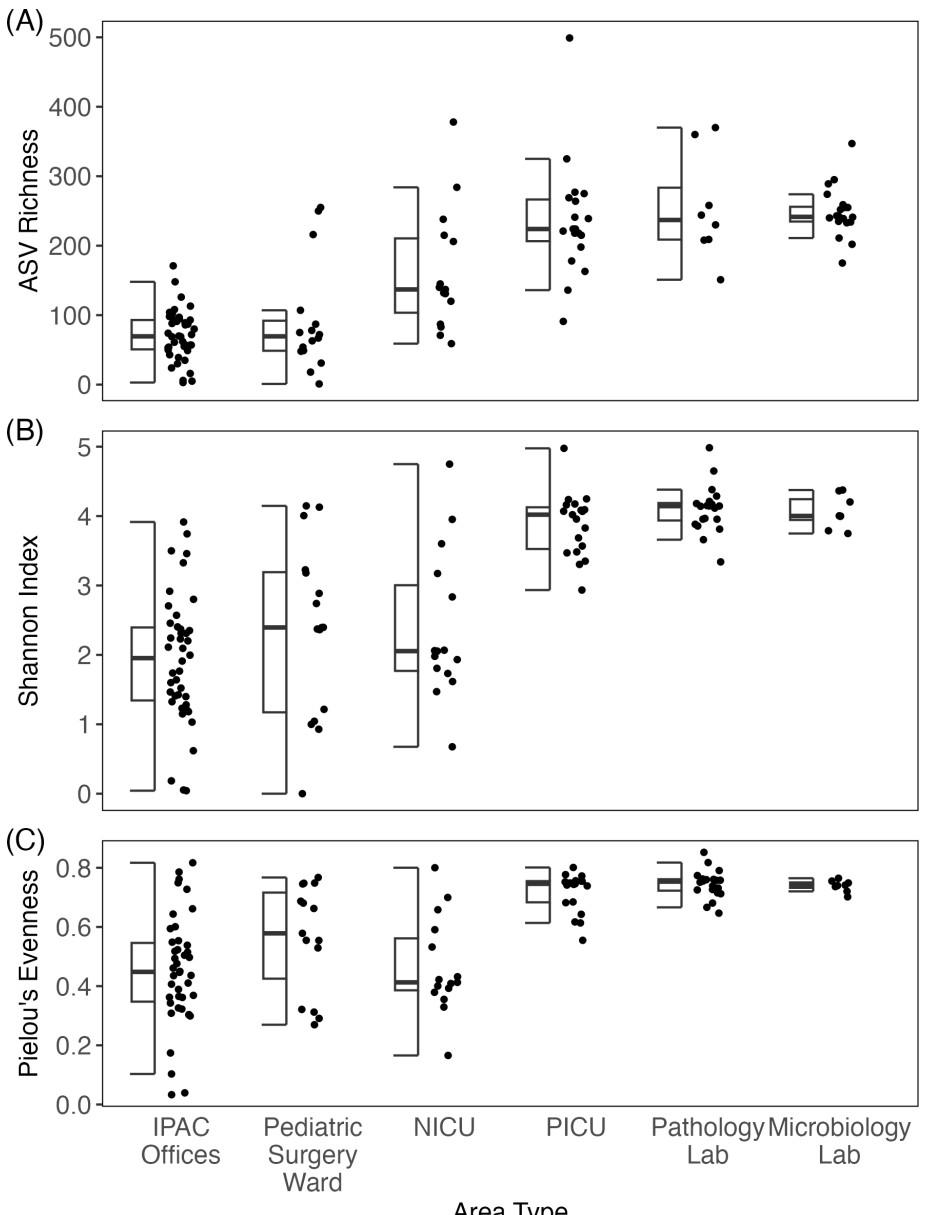

**FIG 2** Alpha diversity in each area type of all door handles, keyboard, and office electronic samples taken after the hospital opened for inpatient care (*n* = 120). These surfaces are high-touch areas of interest and were most widely sampled across all areas. Pathology lab: *n* = 20, IPAC offices: *n* = 42, microbiology lab: *n* = 8, PICU: *n* = 19, NICU: *n* = 15, pediatric surgery ward: *n* = 16. (A) ASV richness, (B) Shannon index, and (C) Pielou's evenness.

PICU, pathology lab, and microbiology lab exhibited the highest diversities. This suggests areas serving different medical activities and cleaning procedures may differ in their sources of microbes and the processes shaping their microbial communities. Surface type differences were not significant (Kruskal-Wallis tests *P*-value > 0.05 for all metrics, Table 1). The overall association between richness and the time after opening was not significant (Pearson = −0.12, *P*-value = 0.199). When examined within each area type, the alpha diversities of the microbial community samples from all areas, except for the pediatric surgery ward, also did not show significant associations with time (Table S4). Although the pediatric surgery ward exhibited significant association, the association was largely produced by one unusual sample. Thus, the time after opening did not exhibit a strong effect on alpha diversity.

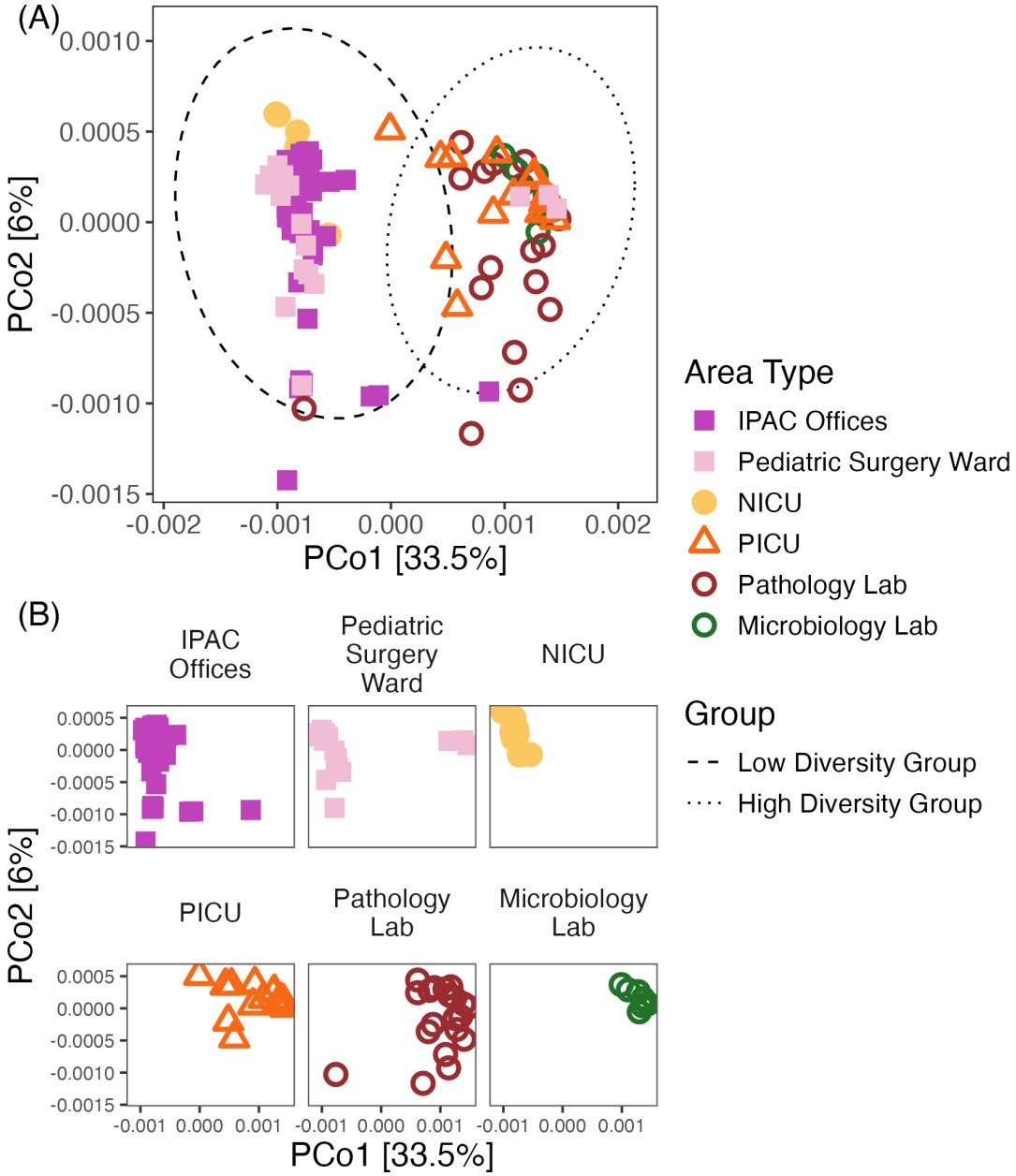

**FIG 3** PCoA with Bray-Curtis dissimilarity of all door handles, keyboard, and office electronic samples taken after the hospital opened for inpatient care ($n = 120$) and color-coded by the area type of where the sample was taken. ASV relative abundances were square-root transformed prior to calculating the dissimilarity. Sample sizes: pathology lab: $n = 20$, IPAC offices: $n = 42$, microbiology lab: $n = 8$, PICU: $n = 19$, NICU: $n = 15$, pediatric surgery ward: $n = 16$. PERMANOVA test: $H_o$: the centroids of the groups are the same. $H_a$: the centroids of the groups are not the same. $P = 0.001$. $H_o$ can be rejected. Solid symbols are area types in the low diversity group, and open symbols are in the high diversity group. (A) Includes multivariate t-distribution ellipses of the low and high diversity groups, and (B) facets of the PCoA by area type.

We analyzed the Bray-Curtis dissimilarities between samples and visualized the results by PCoA. We detected a clear difference in the group centroids, which was associated with the area type (Fig. 3). The samples from the PICU, pathology lab, and microbiology lab formed one cluster, and the samples from NICU, IPAC offices, and pediatric surgery ward formed another cluster. Notably, this separation is associated with trends in alpha diversity, where the microbiology lab, pathology lab, and PICU had higher observed richness and Shannon index values than the other three area types. Specific surface types and time after opening did not visually exhibit strong clustering.

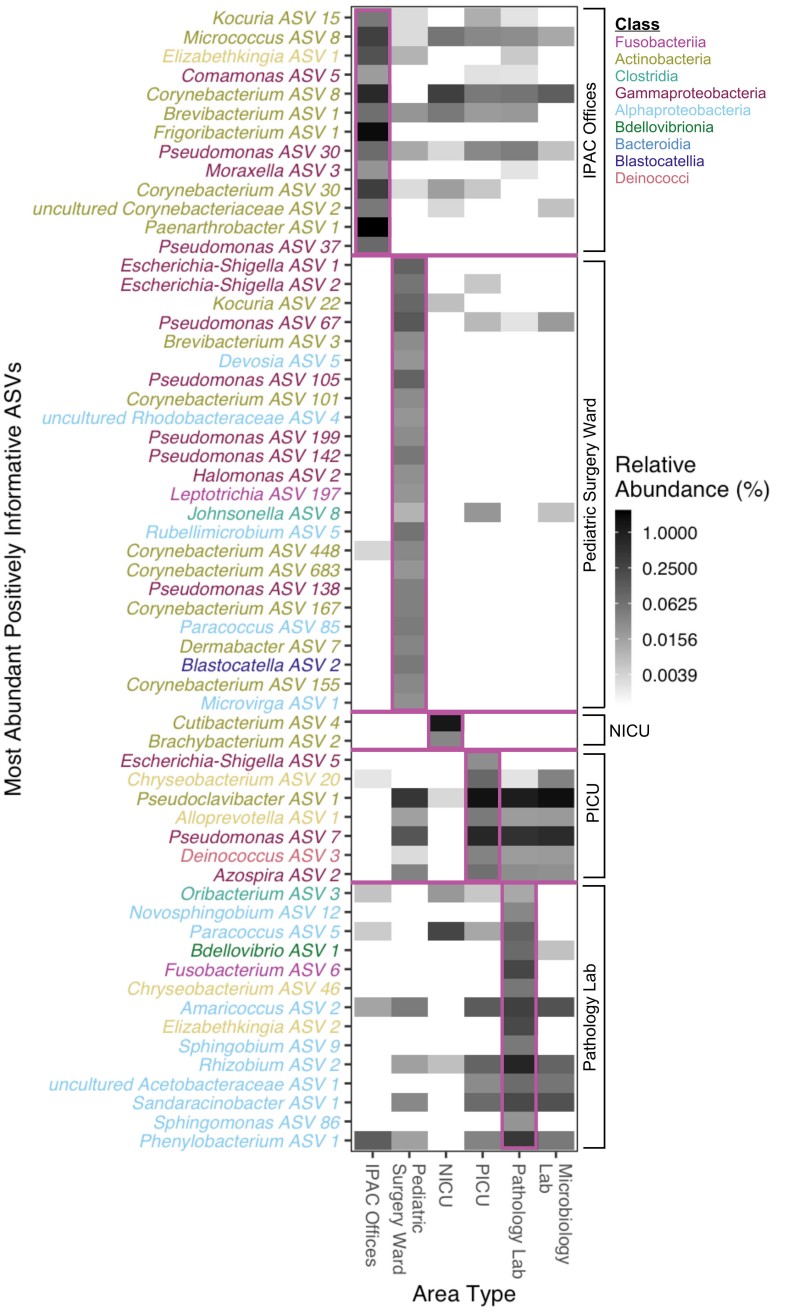

**FIG 4** Heatmap of ASVs identified in the LASSO classifications for area type of origin. Only the ASVs with positive coefficients and cumulative total relative abundances greater than 0.02% are shown. Ten-fold cross-validation repeated 10 times was used to predict if a sample was taken from the area type of interest or another area based on the log-transformed relative abundance of ASVs. Every model, which predicts based on a specific area type, is separated by a line. Within each model, the ASVs are ordered from largest to smallest.

Next, we used the PERMANOVA test to determine how much of the variation in the dissimilarities among samples was explained by area type, surface type, and time after opening in the PERMANOVA test. Area type explained the highest variance (R-squared = 0.301, *P*-value = 0.001; Table 2). Surface type explained much less variance than the area type, although the effect was significant (R-squared = 0.0289, *P*-value = 0.006; Table 2). Further analysis by pairwise PERMANOVA found no significant differences

between surface types (Table S7), and the effect of time after opening was not significant (R-squared = 0.0536, *P*-value = 0.44; Table 2).

## LASSO-classifiers built on informative taxa distinguish area types with high accuracy and outperform SVM and random forest models

LASSO classifications were performed to identify informative ASVs. These are the smallest set of ASVs that distinguish the area types from each other. After repeating 10-fold cross-validation 10 times on each model, 156 ASVs were identified as informative. Figure 4 illustrates the highly abundant and positively associated ASVs, Fig. S6 to S12 shows all informative ASVs for each model, and the minimum value of lambda, the regularization hyperparameter, chosen in each model is reported in Table S8. The testing accuracy for each area type ranged from 86.37% to 100%, indicating that the models performed well (Table 3). The order of magnitude of the LASSO coefficients ranged from $10^{-16}$ to $10^1$, indicating a wide variety in the strength of the association for each ASV (Table S9). Among the informative ASVs, we defined influential ASVs as those with a LASSO coefficient absolute value greater than one, which indicates those ASVs as particularly informative. We identified 18 influential ASVs. An ASV of the genera *Rothia* was associated with the IPAC offices. Two ASVs of *Escherichia-Shigella* were associated with the pediatric surgery ward as were ASVs of *Rhizobium*, *Risungbinella*, *Brevundimonas*, *Kocuria*, and *Deinococcus*. Two ASVs of *Brachybacterium* and an ASV of *Cutibacterium* distinguished the NICU. The most influential ASVs indicative of the PICU were of the genera *Escherichia-Shigella*, and the ASVs associated with the pathology lab were of the genera *Oribacterium*, *Meiothermus*, *Novosphingobium*, *Paracoccus*, *Bdellovibrio*, and *Fusobacterium*. Lastly, no ASVs were found to distinguish samples from the microbiology lab from other samples, likely due to the low number of samples available from that area. Finding a set of features that were distinctive for each area type suggested that the microbial taxa in those environments were temporally stable during the 2 months of sampling.

Of all the informative ASVs, some were present in multiple areas in different abundances, and others were present in only one area, which suggests different dispersal patterns throughout the hospital. Some of these microbes, such as *Leptotrichia*, *Johnsonella*, and *Oribacterium*, could have been sourced from the human oral microbiome, where they are typically found (53, 54). Similarly, there are gut, respiratory, and urogenital microbes, such as *Moraxella* and *Escherichia-Shigella*, that were likely shed by humans onto the surfaces (55, 56). These ASVs may be associated with specific areas because the activities performed either limit or enhance the likelihood of dispersal from different parts of the human body. The outdoor environment is also a potential source of bacteria, and tolerant microbes such as *Blastocatella* and *Deinococus* are indicative of the hospital's desert location (57, 58). LASSO models also identified ASVs of *Kouria*, *Elizabethkingia*, *Comamonas*, and *Chryseobacterium*, and though many of the species in these genera are commensal, a few are emerging pathogens (48, 59–61). 16S sequencing paired with machine learning may be a helpful tool for identifying sources and hot spots of potentially pathogenic organisms.

SVM and random forest models were also employed to distinguish the area types based on their ASV compositions. The tuned hyperparameters for the SVM and random forest models are reported in Table S8. The testing accuracy of these models for each area type ranged widely, from 47.4% to 100% (Table 3). The accuracy of the LASSO models was higher than the class-level accuracy of the SVM and random forest models for all areas, except the pathology lab.

## Culture-based analysis confirmed the presence of antimicrobial-resistant organisms

To identify viable, clinically relevant organisms, we used blood agar to culture all 160 surface microbiome samples collected by the healthcare workers in the PICU, IPAC

**TABLE 3** Machine learning models perform differently in predicting the area type among the post-opening door handle, keyboard, and office electronic samples (*n* = 120)[a]

| Machine learning model | Response variable | Training accuracy | Testing accuracy | Testing MCC |
|---|---|---|---|---|
| **(A) Binary models** | | | | |
| LASSO | IPAC offices | 95.20% | 90.90% | 0.81 |
| LASSO | Pediatric surgery ward | 92.70% | 86.40% | 0 |
| LASSO | NICU | 99.00% | 100.00% | 1 |
| LASSO | PICU | 84.60% | 90.90% | 0.549 |
| LASSO | Pathology lab | 86.20% | 86.40% | 0.5 |
| LASSO | Microbiology lab | 93.00% | 95.50% | 0 |
| **(B) Multiclass models** | | | | |
| SVM linear | Area type | 77.90% | 72.70% | 0.656 |
| Performance by class | | | | |
| | Class: IPAC offices | –[b] | 90.20% | 0.804 |
| | Class: pediatric surgery ward | – | 47.40% | −0.0867 |
| | Class: NICU | – | 100.00% | 1 |
| | Class: PICU | – | 75.40% | 0.417 |
| | Class: pathology lab | – | 100.00% | 1 |
| | Class: microbiology lab | – | 47.60% | −0.0476 |
| SVM radial | Area type | 73.20% | 72.70% | 0.651 |
| Performance by class | | | | |
| | Class: IPAC offices | – | 86.60% | 0.716 |
| | Class: pediatric surgery ward | – | 47.40% | −0.0867 |
| | Class: NICU | – | 100.00% | 1 |
| | Class: PICU | – | 75.40% | 0.417 |
| | Class: pathology lab | – | 100.00% | 1 |
| | Class: microbiology lab | – | 50.00% | 0 |
| Random forest | Area type | 58.70% | 63.60% | 0.539 |
| Performance by class | | | | |
| | Class: IPAC offices | – | 82.10% | 0.629 |
| | Class: pediatric surgery ward | – | 50.00% | 0 |
| | Class: NICU | – | 50.00% | 0 |
| | Class: PICU | – | 75.40% | 0.417 |
| | Class: pathology lab | – | 100.00% | 1 |
| | Class: microbiology lab | – | 50.00% | 0 |

[a]Tuning was performed with 10-fold CV repeated 10 times maximizing the accuracy.
[b]–, not applicable.

offices, pathology lab, microbiology lab, and pediatric surgery ward (no samples were cultured from the NICU). Of the 160 samples, 134 grew colonies. A total of 38 species were identified by MALDI-TOF MS (Fig. 5A). Notably, two organisms on the 2019 CDC Antimicrobial Resistance Threats list were detected: methicillin-resistant *Staphylococcus aureus (MRSA)* and *Acinetobacter parvus* (62). *A. parvus* was not tested for carbapenem resistance and thus is only a potential threat. Four colonies of *MRSA* were cultured from a sample from a keyboard in the microbiology lab, and one colony of *A. parvus* was cultured from a sample from a computer mouse in a PICU patient room.

Since Gram-negative bacteria are more resistant to antibiotics and disinfectants than Gram-positive bacteria, we also specifically noted the number of Gram-positive and Gram-negative species present. Among Gram-positive bacteria, eight species in the genus *Staphylococci* (*S. aureus*, *S. epidermis*, *S.auricularis*, *S. haemolyticus*, *S. succinus*, *S. caprae*, *S. luteus*, and *S. pasteuri*), and five species in the genus *Bacillus* (*B. subitlis*, *B. cereus*, *B. simplex*, *B. pumilus*, and *B. sonorensis*) were detected. Other clinically relevant Gram-positive bacteria detected were *Microccous luteus*, *Brevibacterium casei*, and *B. paucivorans*. Among Gram-negative bacteria, *Pseudomonas stuzeri*, *P. fulva*, and *Pantoea*

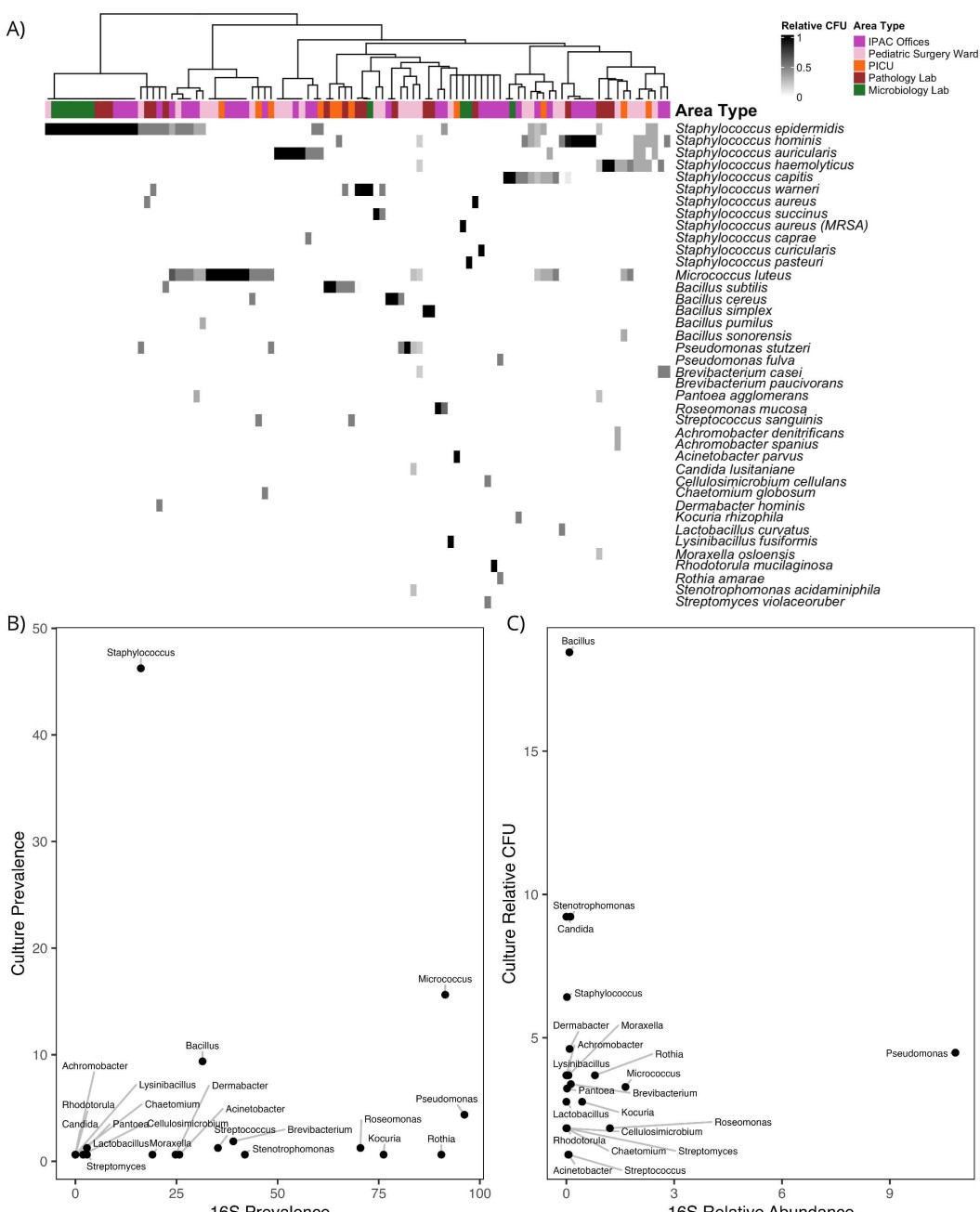

**FIG 5** (A) Heatmap of the relative CFU of each species cultured from all surface samples taken in the IPAC offices, PICU, pathology lab, microbiology lab, and pediatric surgery ward after the hospital opened (*n* = 160). Hierarchical clustering was performed with all samples using the Ward method. The annotation bar indicates the area type of the sample. (B) The prevalences and (C) relative abundances of each genus in each sample as measured by 16S rRNA gene sequencing and as measured by culture analysis. The weak correlation demonstrates both the compatibility of the methods and the added value of 16S sequencing for wider data capture.

*agglomerans* were detected. A majority of these species are harmless and common to the human skin microbiome, however, viable *MRSA* and *A. parvus* in the hospital could put patients at risk.

We compared bacterial culturing and 16S sequencing by examining the prevalence, relative abundance, and clustering data from each method. The prevalences of the genera, as in the proportion of samples the genera were detected in by culturing and 16S sequencing, had a Spearman correlation of 0.239 (*P*-value = 0.012, Fig. 5B).

Similarly, the relative abundances as measured by both methods had a Spearman correlation of 0.134 (*P*-value = 0.163, Fig. 5C). These correlations, though weak, demonstrate the interrelation of the two methods, and highlight how among the limited genera detectable by culturing relative to 16S sequencing there are disparities in representation. For example, *Staphylococcus* was overrepresented, and *Streptococcus*, *Acinetobacter*, and *Pseudomonas* were underrepresented in culturing prevalence. Unlike the PCoA of the 16S data, the hierarchical clustering of samples based on the relative abundance of culturable bacteria did not reflect the area type differences, demonstrating the value of the insight that 16S sequencing provides (Fig. 5A).

## DISCUSSION

The surface microbiomes of hospitals can be influenced by complex factors, but the effects of clinical activities on the surface microbiome remain poorly understood. Thus far, studies have been focused on limited clinical areas in the hospital, such as the intensive care units. By engaging healthcare workers as citizen scientists, our study was able to survey diverse areas of a tertiary hospital where distinct clinical activities were performed and thus operated differently from one another. Healthcare workers chose which high-touch surfaces to sample which increased their awareness of potential microbial reservoirs in their workspace, evidenced by the diverse types of surfaces (Table S3). Potential issues of sampling bias were mitigated by repeated sampling over time and analyzing the most frequently sampled surfaces. We found that clinical activities can be an important factor in shaping the surface microbiome, overriding the effect of surface materials and the operation time. Previous studies have found that the microbiome compositions of the built environment are associated with the occupants as well as the activities performed there, which explains microbiome compositions between different types of buildings, such as a hospital and a museum (2, 63). Here, we see that it may also be the cause of differences within a hospital.

Our sampling approach revealed alpha and beta diversity differences between certain areas. The PICU and NICU are both areas for long-term critical care, yet we found that they exhibited different alpha and beta diversities. A previous study in the surgical subspecialty, hematology, and oncology floors of a hospital showed that patients initially acquire the microbes of the environment and then spread their microbes (13). If we consider the healthcare workers as consistent occupants of the hospital who practice stringent infection control measures, we can assume variability between areas to be caused by differences in patient occupancy. Potentially, the microbes in the PICU are exchanged at a higher rate than those in the NICU because it continually receives patients from outside of the hospital, whereas the neonates in the NICU have never left the hospital environment and thus do not bring in external microbes. The pediatric surgery ward, a short-term care unit, exhibited low alpha diversity, possibly due to stringent cleaning protocols and a shorter patient occupancy than the PICU. The alpha and beta diversities of the pediatric surgery ward were similar to those of the IPAC offices, which are occupied only by the IPAC workers. In the case of the offices, the low alpha diversity may be attributed to restricted access.

Access is not the only factor in shaping surface microbial diversity. Viewed under the community assembly perspective, a hospital unit can be viewed as an island, and the entire hospital can be viewed as islands connected by various dispersal routes, including passive vectors (64, 65). We found that the microbiology lab and pathology lab had similar alpha and beta diversities, and the alpha diversities of these areas were significantly higher than all other areas. While they can be accessed only by restricted personnel, they can be seeded by biological samples collected from patients throughout the hospital. Therefore, dispersals mediated by passive vectors play an important role in seeding a local surface microbiome. Thus, infection control strategies targeting passive vectors in and out of the laboratories can be particularly important.

Conversely, our temporal analysis did not find alpha or beta diversity differences between samples taken in the NICU patient rooms before and after the hospital was

operational. This result is contrary to other reports of a significant shift in the micro-biome composition in a newly built hospital (13). The stability of the microbiome in our study may be due to the restricted access to the NICU. As discussed, the NICU provides long-term care, and we have seen that its alpha diversity remained relatively low even after the hospital opened. Restricting NICU access to healthcare workers and neonatal patients limited the potential for major sources of new microbes, and thus the microbiome remained stable.

Since variations in microbial diversity could be partially explained by the area type, we conducted further analyses to understand the compositional microbiome differences between hospital areas. Machine learning modeling, which has often been applied to microbiome data, can identify microbial indicators of any response of interest. In previous studies, the samples have typically been of human microbiomes, and the response is a health outcome or diagnosis, such as colon lesions, diabetes, or smoking habits (18, 19, 66, 67). Robust models have been generated in a health context, and these same types of models could also be applied to the environmental microbiome for either predictive or explanatory analysis (16, 19, 68). Utilizing both environmental and human microbiomes, Marotz et al. predicted the likelihood of detecting SARS-CoV-2 RNA in surface samples of hospital floors, nares, skin, and stools and identified bacteria that were likely associated with the virus. Using LASSO classifications in an explanatory analysis, we identified several ASVs that are characteristic of each area. These statistical results point even more strongly to the unique microbiome signatures created in each area, likely by the inherent differences in the occupants and activities each area houses. For example, ASVs of *Escherichia-Shigella*, *Leptotrichia*, and *Johnsonella* were biomarkers of the pediatric surgery ward which indicates higher shedding of these gut and oral microbes. These microbes may originate from patients who have recently undergone surgery and shed them, such as from incisions or incubation tubes. Identification of these biomarkers and dispersal patterns could aid in infection control monitoring by specifying which bacteria to target in culturing methods and which areas to inspect.

From a methodological perspective, SVM and random forest models yielded lower accuracies than LASSO models for each area type, except the Pathology Lab. Unlike LASSO classification, these more complex machine learning models can identify multiple classes. Due to this complexity, the computational time is longer and post hoc analysis is needed to identify important features. Our findings demonstrate that the simpler LASSO is the optimal model for our purpose of identifying informative ASVs from our specific data set. LASSO is easier to implement and interpret, and as demonstrated, the identified biomarkers are useful for the surveillance of different sources and dispersal patterns.

Culture-based analysis and sequencing-based analysis revealed different aspects. Culture-based analysis confirmed the prevalence and viability of clinically relevant species, including antibiotic-resistant organisms. At the same time, the 16S sequencing data analysis revealed clustering by area that was not identified by the culturing data analysis. Additionally, machine learning methods were applied to obtain even further information from the data. We could conduct these analyses due to the many additional taxa identified by 16S sequencing, a finding consistent with other hospital studies and expected since very few environmental bacteria are viable and culturable (10). Even among the genera identified by both methods, culturing detected them disproportion-ately. This result is expected considering the selective agar medium and differential bacterial growth rates. When surveilling a hospital microbiome, both analyses should be used for a comprehensive picture. Advanced metagenomic sequencing could also be applied, however, we demonstrated that 16S sequencing remains a useful tool, particularly in low biomass and large sampling studies.

Infection control approaches should consider the area of the hospital being investigated. We have seen how distinct microbiomes are created in different areas, possibly due to differences in occupants, activities, and access restrictions. These differences could translate to differences in the potential for pathogenic and antibiotic-resistant organisms to grow and be dispersed. For example, if an antibiotic-resistant

organism is detected in one area of the hospital, we would anticipate a higher likelihood of its presence in areas with similar patterns of diversity and dispersal, and we could implement surveillance and infection control accordingly. Furthermore, longitudinal studies should be done to determine if this compartmentalization is evident in other large hospitals and if the microbial communities are stable over time. If the microbial dispersal patterns and mechanisms are better understood, they can be controlled to reduce infection risk and disease transmission.

## ACKNOWLEDGMENTS

We thank Sidra Medicine for the support provided through a grant (SDR-200032) and the McKelvey School of Engineering at Washington University in St. Louis for the support provided by the McKelvey Faculty Startup Fund. K.C. is supported by the National Science Foundation (NSF) Graduate Research Fellowship Program (Grant No. DGE 2139839).

## AUTHOR AFFILIATIONS

[1]Department of Energy, Environmental and Chemical Engineering, Washington University in St. Louis, St. Louis, Missouri, USA

[2]Department of Pathology, Sidra Medicine, Doha, Qatar

[3]Department of Pathology and Laboratory Medicine, Weill Cornell Medicine-Qatar, Doha, Qatar

[4]Faculty of Medicine, University of British Columbia, Vancouver, Canada

[5]Infectious Diseases Research Laboratory, National Centre for Infectious Diseases, Singapore

[6]Division of Biological and Biomedical Sciences, Washington University in St. Louis, St. Louis, Missouri, USA

[7]Department of Computer Science and Engineering, Washington University in St. Louis, St. Louis, Missouri, USA

## PRESENT ADDRESS

Fangqiong Ling, Department of Energy, Environmental and Chemical Engineering, Washington University in St. Louis, St. Louis, Missouri, USA

## AUTHOR ORCIDs

Kaseba Chibwe  http://orcid.org/0000-0003-4290-7025
Lin Zhang  http://orcid.org/0000-0001-9115-6585
Clement Tsui  http://orcid.org/0000-0001-5129-1037
Patrick Tang  http://orcid.org/0000-0003-1583-5484
Fangqiong Ling  http://orcid.org/0000-0003-1546-5647

## FUNDING

| Funder | Grant(s) | Author(s) |
| --- | --- | --- |
| Sidra Medicine (سدرة للطب) | SDR-200032 | Sathyavathi Sundararaju |
|  |  | Clement Tsui |
|  |  | Patrick Tang |
| WashU \| McKelvey School of Engineering (McKelvey Engineering) |  | Kaseba Chibwe |
|  |  | Lin Zhang |
|  |  | Fangqiong Ling |
| NSF \| National Science Foundation Graduate Research Fellowship Program (GRFP) | DGE 2139839 | Kaseba Chibwe |

## AUTHOR CONTRIBUTIONS

Kaseba Chibwe, Data curation, Formal analysis, Methodology, Software, Visualization, Writing – original draft, Writing – review and editing | Sathyavathi Sundararaju, Conceptualization, Data curation, Investigation, Methodology, Validation, Writing – original draft, Writing – review and editing | Lin Zhang, Data curation, Writing – review and editing | Clement Tsui, Conceptualization, Investigation, Supervision, Writing – review and editing | Patrick Tang, Conceptualization, Funding acquisition, Methodology, Project administration, Resources, Supervision, Writing – review and editing | Fang-qiong Ling, Conceptualization, Funding acquisition, Project administration, Resources, Supervision, Writing – original draft, Writing – review and editing

## DATA AVAILABILITY

Raw sequence data are available in the NCBI Sequence Read Archive under Bioproject ID PRJNA1043714. The data and the code used to generate the results are available at https://github.com/linglab-washu/Sidra-Hospital-Microbiome.

## ADDITIONAL FILES

The following material is available online.

### Supplemental Material

**Supplemental figures (Spectrum00296-24-S0001.docx).** Fig. S1 to S12.
**Supplemental tables (Spectrum00296-24-S0002.docx).** Tables S1 to S10.

### Open Peer Review

**PEER REVIEW HISTORY (review-history.pdf).** An accounting of the reviewer comments and feedback.

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
