## [Reviewer comments · Microbiology Spectrum]

Microbiology Spectrum

Intra-hospital Microbiome Variability Is Driven by Accessibility and Clinical Activities

Kaseba Chibwe, Sathyavathi Sundararaju, Lin Zhang, Clement Tsui, Patrick Tang, and Fangqiong Ling

Corresponding Author(s): Fangqiong Ling, Washington University in St Louis

Review Timeline:

Submission Date:	February 8, 2024
Editorial Decision:	March 22, 2024
Revision Received:	May 28, 2024
Accepted:	May 30, 2024

Editor: Jan Claesen

Reviewer(s): Disclosure of reviewer identity is with reference to reviewer comments included in decision letter(s). The following individuals involved in review of your submission have agreed to reveal their identity: Ben Ma (Reviewer #2); Xinzhao Tong (Reviewer #3)

Transaction Report:

DOI: <https://doi.org/10.1128/spectrum.00296-24>

Re: Spectrum00296-24 (Intra-hospital Microbiome Variability Is Driven by Accessibility and Clinical Activities)

Dear Dr. Fangqiong Ling:

Thank you for the privilege of reviewing your work. Below you will find my comments, instructions from the Spectrum editorial office, and the reviewer comments.

Thanks for submitting your research to Spectrum. Your manuscript has now been evaluated by three independent Reviewers whom are overall enthusiastic about your work (as am I). The Reviewers have brought up some comments and suggestions to improve your manuscript and I would be happy to consider a revised version which addresses these comments in a point-by-point manner.

Revision Guidelines

Sincerely,
Jan Claesen
Editor
Microbiology Spectrum

Reviewer #1 (Comments for the Author):

The paper written by Chibwe et al., uses a combination of deep-amplicon sequencing and culture-dependent approaches to identify microbial spatial patterns for prevention and control in nosocomial settings. Very few times, as clinician or researcher we

have the opportunity to set a baseline for a hospital from a microbiological perspective. I consider this paper to be very well written and the results seem solid. My comments described below are small in nature:

In some parts of the paper Pearson was used over Spearman and viceversa. Was there a specific reason for this?. In this context line 267, presents Pearson- $\rho=0.287$ with a significant value of p-value $<2.2e-16$. While significant, that positive value Pearson- $\rho=0.287$ seems weak.

There is a typo in line 163, it should be storage.

Line 149, the fact "the volunteers sampled five surface sites of their choice" seems biased. Could you elaborate more on this experimental design?.

Line 339, besides Principal Coordinate Analysis reported here, was any other model ran?. Regarding PCoA, axis in this figure seems to indicate principal component analysis (PC1 vs PC2). I would use PCo1 vs PCo2.

Lines 500-501, antibiotic resistant organisms were identified. Was a MIC conducted during any point of the study (i.e. baseline and April-May 2018).

Reviewer #2 (Comments for the Author):

General comment:

In this manuscript, the hospital environmental microbiome was investigated by collecting surface samples across different areas in a hospital and assessed using both 16S rRNA gene sequencing and culture-based methods. The diversity and microbial composition was analyzed and machine learning models were used to identify distinct members in sample collected from different areas. The manuscript is well-written and easy to follow and the result presentation is well-organized. The topic of this manuscript should be of interest to the readers of Microbiology Spectrum. However, I would like the authors to address the following comments before I can recommend this manuscript for publication.

Sample collection:

Is there an instruction for selecting sampling locations for the healthcare workers or they can collect samples from wherever they think are high-touch areas? Considering these areas may also frequently cleaned/sanitized, it would be great to discuss what the criteria is for selecting sampling locations for microbiome monitoring.

ML:

More justification is needed for using Machine Learning in this study. ML is used to identify the distinct bacterial members in different areas, which can also be achieved via simply comparing the relative abundance/occupancy across different sampling areas using R or other statistical software. It is unclear why ML is needed here and how it is better than simply comparing the data.

Also, it is unclear what the ecological and practical importance of identifying distinct members in different areas is. The authors should improve their discussion on why we want to understand this.

Several terms are defined and used in Line 350-392 and it is a bit confusing. What are the criteria for members of the distinguishing set? Are they also referred as "informative ASVs"? Are "influential ASVs" just "informative ASVs" with greater ML coefficients?

Finally, the authors compared LASSO, SVM, and forest models and concluded that "LASSO is the optimal model for identifying informative ASVs" (Line 496). I think this conclusion may be biased considering only limited number of samples were used for training the models ($n=120$) and different classes were used as the response variable for LASSO and the other two models (binary vs. multiclass). I would recommend toning down such claim and focusing on conclusions that specifically related to the experimental setup in this study.

Culture-based analysis:

It is unclear why the authors wanted to compare the sequencing based results vs. culture-based results. The culture-based analysis is highly dependent on the culture medium selection. In this manuscript, the authors selected sheep blood agar (Line 175) but no justification was provided. It would be great to provide some justification and descriptions of previous studies using similar methods here.

The results basically suggest that there is weak to no significant correlation (P-values are needed) on the prevalence (need definition in the manuscript) and relative abundance between sequencing results and cell culture results (Line 419). This is expected considering only very small portions of the bacteria may grow on the agar plates and even the ones grow may have

very different growth rate, which may affect the relative abundance (assuming the relative abundance here are after growth on agar plates).

Reviewer #3 (Comments for the Author):

Keseba et al. conducted an extensive study on the surface bacterial communities in various areas of a hospital, both before and after its opening. They utilized amplicon sequencing and culture methods for analysis. Overall, the manuscript is well-written and has a logical structure. One major suggestion for improvement is to emphasize the pathogen dynamics in the sequencing data analysis. This includes aspects such as pathogen diversity, intra-sample compositional differences of pathogens, and machine-learning predictions of area-specific pathogen markers. Such enhancements could significantly elevate the manuscript's significance by reducing the emphasis on indoor environmental and human commensal taxa.

Introduction

L71, "but also commensal microorganisms, which are normally non-pathogenic," suggests to be changed into "but also opportunistic microorganisms, which are normally non-pathogenic to healthy individuals". Because there are rare cases in which non-pathogenic and commensal microorganisms can cause diseases.

L76-85, suggests deleting, as the 16S amplicon sequencing is widely used, and readers in this field should have a good understanding of this technology.

L117, replace "microbiome community" with "microbial community".

L172, why did you use the water control? Why didn't you include an unused swab also as a control?

L183, sequence quality score above 10? To be honest, 10 is very low ... please indicate the average score of sequencing you obtained after quality filtering.

L191 suggests using greengene2 for taxonomic annotation, which was released recently and has a higher resolution at fine levels.

L198 sampling date is also one of the metadata.

L219, why didn't you apply weighted and unweighted unifracs distance metrics, which were widely used in the beta diversity of 16S amplicon data analysis?

L219-228, in the statistics of permanova, because you have multiple factors, how did you control the influence of others when you were testing the significance of one factor? Please illustrate clearly in the method section. It is good that you converted the categorical factor "sampling data" to a numeric factor "number of days after opening" in statistics.

L246-261 suggests moving to the method section.

L268, is there a reference for the core microbiome at a prevalence cutoff of 40%? In addition to the prevalence, the relative abundance is also important for defining the core microbiome; some previous studies have suggested using 1% as the cutoff for the core taxa, if I remember correctly.

L270-291, among such many areas, which area and which sample types contain the highest relative abundance of pathogens? It suggests using a heat map showing the results.

Table 1 and table 2 suggest to be put as supplementary files.

Discussion

L448, too absolute, the healthcare workers can bring in external microbes in the NICU.

Point-by-Point Response to Reviewers [Paper #Spectrum00296-24]

TITLE: Intra-hospital Microbiome Variability Is Driven by Accessibility and Clinical Activities

AUTHORS: Kaseba Chibwe, Sathyavathi Sundararaju, Lin Zhang, Clement Tsui, Patrick Tang, and Fangqiong Ling

The authors thank the reviewers for the comments and feedback on our manuscript. Below we have addressed each comment. The reviewers' comments are *italicized*, our responses are normal text, and direct references to changes in the manuscript are **highlighted**.

R1: Reviewer #1 (Comments for the Author):

GC1: *The paper written by Chibwe et al., uses a combination of deep-amplicon sequencing and culture-dependent approaches to identify microbial spatial patterns for prevention and control in nosocomial settings. Very few times, as clinician or researcher we have the opportunity to set a baseline for a hospital from a microbiological perspective. I consider this paper to be very well written and the results seem solid. My comments described below are small in nature:*

Response: We thank the reviewer for the positive feedback. We have carefully considered all the comments from the reviewer and found them helpful. Revisions and clarifications based on the reviewer's suggestions and questions, respectively, are provided under each comment.

GC2: *In some parts of the paper Pearson was used over Spearman and vice versa. Was there a specific reason for this?*

Response: Thank you for bringing this to our attention. We have changed the key correlations to be consistently determined by Spearman tests since there were no assumptions of linearity for the relationships analyzed.

Line 266-267: "The relative abundances and prevalences of the ASVs exhibited positive correlation (p-value <2.2e-16, **Spearman- ρ = 0.672**; Figure S2-S3); therefore, the core-satellite theory was applied."

SC1: *In this context line 267, presents Pearson- ρ = 0.287 with a significant value of p-value <2.2e-16. While significant, that positive value Pearson- ρ = 0.287 seems weak.*

Response: We thank the reviewer for the thoughtful comment. The core-satellite theory is based on the observation of a positive correlation. After extensive literature research, we did not find a specified strength of the correlation in the original paper by Hanski et.

al or studies which cite the theory. We did find that abundance has a nonlinear relationship with prevalence. Therefore, we have changed to a Spearman correlation instead to avoid assumptions of linearity.

Line 266-267: “The relative abundances and prevalences of the ASVs exhibited positive correlation (p-value <2.2e-16, Spearman- ρ = 0.672; Figure S2-S3); therefore, the core-satellite theory was applied.”

Reference: Hanski, I. Dynamics of Regional Distribution: The Core and Satellite Species Hypothesis. *Oikos* **1982**, 38 (2), 210. <https://doi.org/10.2307/3544021>.

SC2: *There is a typo in line 163, it should be storage.*

Response: Thank you for the correction. The sentence is now, Line 156-158: “For samples collected by healthcare workers, 100 μ L of the eluent was plated on sheep blood agars (MicroLab Medical, Qatar) prior to storage.”

SC3: *Line 149, the fact "the volunteers sampled five surface sites of their choice" seems biased. Could you elaborate more on this experimental design?*

Response: This point was also raised by Reviewer 2. The reviewer’s comment helped us see the need to better clarify the justifications of our approach. To encourage awareness and compliance with hospital infection control policies, volunteers were given an orientation to the study, examples of high-touch surfaces in the hospital, and training in sample collection. We agree that there may be bias in the choice of surface sampled, but we feel that this was balanced by the benefit that the volunteers would be better at identifying high-touch surfaces in their own workplace, and that we had serial samples from the same sites at the same time of day over the study period. To bring this point to light we’ve added to the Discussion:

Line 430-434: “Healthcare workers chose which high-touch surfaces to sample which increased their awareness of potential microbial reservoirs in their workspace, evidenced by the diverse types of surfaces (Table S3). Potential issues of sampling bias were mitigated by repeated sampling over time and analyzing the most frequently sampled surfaces.”

SC4: Line 339, besides Principal Coordinate Analysis reported here, was any other model run? Regarding PCoA, the axis in this figure seems to indicate principal component analysis (PC1 vs PC2). I would use PCo1 vs PCo2.

Response: We've added an NMDS plot to assess the Bray-Curtis distances (Figure S7). In the PCoA plots, the axes titles have been changed to PCo1 vs PCo2.

FIG S7 (R1.SC4) NMDS plots based on Bray-Curtis distance of all door handle, keyboard, and office electronic samples taken after the hospital opened for inpatient care ($n = 120$) and color coded by the area type of where the sample was taken including multivariate t distribution ellipses of the low and high diversity groups. Sample sizes: Pathology lab: $n = 20$, IPAC offices: $n = 42$, Microbiology lab: $n = 8$, PICU: $n = 19$, NICU: $n = 15$, Pediatric surgery ward: $n = 16$. PERMANOVA test: H_0 : The centroids of the groups are the same. H_a : The centroids of the groups are not the same. $p = 0.001$. H_0 can be rejected. Solid symbols are area types in the low diversity group, and open symbols are in the high diversity group.

A) NMDS completed with all samples and three dimensions, showing the second and third axes, and B) NMDS with one outlier sample removed and two dimensions. ASV relative abundances were square-root transformed prior to calculating the Bray-Curtis dissimilarity.

SC5: *Lines 500-501, antibiotic resistant organisms were identified. Was a MIC conducted during any point of the study (i.e. baseline and April-May 2018).*

Response: We identified MRSA by inoculating all *S. aureus* isolates onto chromogenic medium for MRSA detection. We agree with the reviewer that MIC tests on AMR positive strains will be important follow-up work on this study. Unfortunately, antibiotic resistance testing was not within the scope and funding of this project. As indicated in the manuscript, we also identified *Acinetobacter parvus* but we did not conduct phenotypic testing for carbapenem resistance.

R2: Reviewer #2 (Comments for the Author):

General comment:

GC1: *In this manuscript, the hospital environmental microbiome was investigated by collecting surface samples across different areas in a hospital and assessed using both 16S rRNA gene sequencing and culture-based methods. The diversity and microbial composition was analyzed and machine learning models were used to identify distinct members in samples collected from different areas. The manuscript is well-written and easy to follow and the result presentation is well-organized. The topic of this manuscript should be of interest to the readers of Microbiology Spectrum. However, I would like the authors to address the following comments before I can recommend this manuscript for publication.*

Response: We thank the reviewer for the overall positive feedback. We found the constructive comments helpful. We have carefully considered all the comments raised by the reviewer. Revisions, additional analyses, and clarifications have been made. We believe that these revisions following the reviewer's suggestions have made the manuscript more robust.

Sample collection:

GC2: *Is there an instruction for selecting sampling locations for the healthcare workers or they can collect samples from wherever they think are high-touch areas? Considering these areas may also be frequently cleaned/sanitized, it would be great to discuss what the criteria is for selecting sampling locations for microbiome monitoring.*

Response: This point was also raised by Reviewer 1. The reviewer's comment helped us see the need to better clarify the justifications of our approach. To encourage awareness and compliance with hospital infection control policies, volunteers were given an orientation to the study, examples of high-touch surfaces in the hospital, and training in sample collection. We agree that there may be bias in the choice of surface sampled, but we feel that this was balanced by the benefit that the volunteers would be better at identifying high-touch surfaces in their own workplace, and that we had serial samples from the same sites at the same time of day over the study period. To bring this point to light we've added to the Discussion:

Line 430-434: "Healthcare workers chose which high-touch surfaces to sample which increased their awareness of potential microbial reservoirs in their workspace, evidenced by the diverse types of surfaces (Table S3). Potential issues of sampling bias were mitigated by repeated sampling over time and analyzing the most frequently sampled surfaces."

ML:

GC3: *More justification is needed for using Machine Learning in this study. ML is used to identify the distinct bacterial members in different areas, which can also be achieved via simply comparing the relative abundance/occupancy across different sampling areas using R or other statistical software. It is unclear why ML is needed here and how it is better than simply comparing the data.*

Response: While statistical methods can be used to identify differential bacteria, machine learning models better account for variability introduced by the complex interdependent relationships between different bacteria in the microbiome and identify a robust set of features through cross-validation. The reviewer's comment helped us see the need to better clarify the justifications for our approach. We have added the following to the introduction:

Line 90-96: "The application of machine learning to 16S rRNA sequencing data can identify biomarkers through feature selection. Machine learning models account for inherent complexity and variation within the microbiome which would be otherwise overlooked by traditional statistical methods. The identified biomarkers can be potential pathogen-associated sensors, such as the association between *Rothia* sp. and SARS-CoV-2 virus found in another hospital study. These types of cross-validated models can be integrated with infection control and should be further explored."

GC4: *Also, it is unclear what the ecological and practical importance of identifying distinct members in different areas is. The authors should improve their discussion on why we want to understand this.*

Response: In our study, the models are used for their explanatory value and identify key ASVs which are different among the areas. There is also future potential for similar models to be implemented in infection control and surveillance. The reviewer's comment helped us see the need to better clarify the potential applications of the biomarker results. The following has been added to the Discussion:

Line 487-496: "Using LASSO classifications in an explanatory analysis, we identified several ASVs that are characteristic for each area. These statistical results point even more strongly to the unique microbiome signatures created in each area, likely by the inherent differences in the occupants and activities each area houses. For example, ASVs of *Escherichia-Shigella*, *Leptotrichia*, and *Johnsonella* were biomarkers of the pediatric

surgery ward which indicates higher shedding of these gut and oral microbes. These microbes may originate from patients that have recently undergone surgery and shed them, such as from incisions or incubation tubes. Identification of these biomarkers and dispersal patterns could aid in infection control monitoring by specifying which bacteria to target in culturing methods and which areas to inspect.”

SC1: *Several terms are defined and used in Line 350-392 and it is a bit confusing. What are the criteria for members of the distinguishing set? Are they also referred to as "informative ASVs"? Are "influential ASVs" just "informative ASVs" with greater ML coefficients?*

Response: A few sentences were changed to clarify informative and influential ASVs. They read,

Line 346-357: “LASSO classifications were performed to identify informative ASVs. These are the smallest set of ASVs that distinguishes the area types from each other. After repeating 10-fold cross-validation ten times on each model, 156 ASVs were identified as informative. [...] Among the informative ASVs, we defined influential ASVs as those with a LASSO coefficient absolute value greater than one, which indicates those ASVs as particularly informative. We identified 18 influential ASVs.”

SC2: *Finally, the authors compared LASSO, SVM, and forest models and concluded that "LASSO is the optimal model for identifying informative ASVs" (Line 496). I think this conclusion may be biased considering only limited number of samples were used for training the models (n=120) and different classes were used as the response variable for LASSO and the other two models (binary vs. multiclass). I would recommend toning down such claim and focusing on conclusions that specifically related to the experimental setup in this study.*

Response: We thank the reviewer for the thoughtful comment. Additional clarifications have been made to describe the approach and refine the interpretation of the results. In fact, though the models vary in being binary or multiclass, we examined the accuracy on a class-by-class basis. To clarify this, the sentence was changed to:

Line 497-498: “From a methodological perspective, SVM and random forest models yielded lower accuracies than LASSO models for each area type, with the exception of the Pathology Lab.”

We also intend to emphasize that LASSO performed best for our particular case, not generally. In order to focus on this aspect in our conclusion, we specify:

Line 501-503: “Our findings demonstrate that the simpler LASSO is the optimal model for our purpose of identifying informative ASVs from our specific dataset.”

Last but not least, Table 3 has also been revised to more clearly distinguish binary and multiclass model accuracies.

Table 3. Machine learning models perform differently predicting the area type among the post-opening door handle, keyboard, and office electronic samples (n = 120).¹

Machine Model	Learning	Response Variable	Training Accuracy	Testing Accuracy	Testing MCC
A) Binary Models					
LASSO		IPAC Offices	95.2%	90.9%	0.810
LASSO		Pediatric Surgery Ward	92.7%	86.4%	0
LASSO		NICU	99.0%	100.0%	1
LASSO		PICU	84.6%	90.9%	0.549
LASSO		Pathology Lab	86.2%	86.4%	0.500
LASSO		Microbiology Lab	93.0%	95.5%	0
B) Multiclass Models					
SVM Linear		Area Type	77.9%	72.7%	0.656
Performance By Class					
		Class: IPAC Offices	-	90.2%	0.804
		Class: Pediatric Surgery Ward	-	47.4%	-0.0867
		Class: NICU	-	100.0%	1
		Class: PICU	-	75.4%	0.417
		Class: Pathology Lab	-	100.0%	1
		Class: Microbiology Lab	-	47.6%	-0.0476
SVM Radial		Area Type	73.2%	72.7%	0.651
Performance By Class					
		Class: IPAC Offices	-	86.6%	0.716
		Class: Pediatric Surgery Ward	-	47.4%	-0.0867
		Class: NICU	-	100.0%	1
		Class: PICU	-	75.4%	0.417
		Class: Pathology Lab	-	100.0%	1
		Class: Microbiology Lab	-	50.0%	0
Random Forest		Area Type	58.7%	63.6%	0.539
Performance By Class					
		Class: IPAC Offices	-	82.1%	0.629
		Class: Pediatric Surgery Ward	-	50.0%	0
		Class: NICU	-	50.0%	0
		Class: PICU	-	75.4%	0.417
		Class: Pathology Lab	-	100.0%	1
		Class: Microbiology Lab	-	50.0%	0

¹Tuning was performed with 10-fold CV repeated 10 times maximizing the accuracy.

Culture-based analysis:

GC5: *It is unclear why the authors wanted to compare the sequencing based results vs. culture-based results. The culture-based analysis is highly dependent on the culture medium selection. In this manuscript, the authors selected sheep blood agar (Line 175) but no justification was provided. It would be great to provide some justification and descriptions of previous studies using similar methods here.*

Response: We agree that culture-based results are highly biased based on the medium used. We chose sheep blood agar as that was the most cost-effective medium to identify the majority of clinically-relevant bacteria in the hospital. One of the primary reasons for including bacterial culture in our study was to prove that there were viable bacteria on these surfaces as one of the questions from clinician stakeholders is whether the 16S results have any clinical relevance for patients or for hospital infection control. The comparison between sequencing and culture was to highlight the additional value of sequencing over culture for microbial community structure characterization. We have added to the sentence to the Methods to add appropriate justification:

Line 169-170: “Sheep blood agar was the cost-effective medium that would identify most clinically-relevant bacteria.”

SC3: *The results basically suggest that there is weak to no significant correlation (P-values are needed) on the prevalence (need definition in the manuscript) and relative abundance between sequencing results and cell culture results (Line 419). This is expected considering only very small portions of the bacteria may grow on the agar plates and even the ones that grow may have very different growth rates, which may affect the relative abundance (assuming the relative abundance here is after growth on agar plates).*

Response: We agree that the result is as expected. We’ve added the following to the Discussion to highlight this point:

Line 513-515: “Even among the genera identified by both methods, culturing detected them disproportionately. This result is expected considering the selective agar medium and differential bacterial growth rates.”

We have also clarified the definition of prevalence:

Line 412-414: “The prevalences of the genera, as in the proportion of samples the genera was detected in by culturing and 16S sequencing, had a Spearman correlation of 0.239 (p-value = 0.012, Figure 5B).”

R3: Reviewer #3 (Comments for the Author):

GC1: *Kaseba et al. conducted an extensive study on the surface bacterial communities in various areas of a hospital, both before and after its opening. They utilized amplicon sequencing and culture methods for analysis. Overall, the manuscript is well-written and has a logical structure. One major suggestion for improvement is to emphasize the pathogen dynamics in the sequencing data analysis. This includes aspects such as pathogen diversity, intra-sample compositional differences of pathogens, and machine-learning predictions of area-specific pathogen markers. Such enhancements could significantly elevate the manuscript's significance by reducing the emphasis on indoor environmental and human commensal taxa.*

Response: We thank the reviewer for the overall positive feedback. The constructive comments raised by the reviewer helped us see places where revisions can help improve the clarity and robustness of the manuscript. Extensive revisions have been made to the manuscript, including additional analyses, visualization, and revisions of texts. We believe that the revised manuscript has improved in its robustness.

Introduction

SC1: *L71, "but also commensal microorganisms, which are normally non-pathogenic," suggests to be changed into "but also opportunistic microorganisms, which are normally non-pathogenic to healthy individuals". Because there are rare cases in which non-pathogenic and commensal microorganisms can cause diseases.*

Response: Thank you for the suggestion. The sentence now reads:

Line 68-71: “Hospitalized patients often have comorbidities or immunocompromising conditions that make them more susceptible to serious infections from not only pathogenic microorganisms, but also opportunistic microorganisms, which are normally non-pathogenic to healthy individuals.”

SC2: *L76-85, suggests deleting, as the 16S amplicon sequencing is widely used, and readers in this field should have a good understanding of this technology.*

Response: The reviewer’s comment helped us see the need to provide more specific information here. This paragraph and the citations used are meant to specifically anchor the use of 16S amplicon sequencing in hospital microbiomes. This part of the literature review has been succinctly summarized in the following way:

Line 76-78: “Temporal and spatial dynamics in the hospital environment have been previously investigated using culture-dependent and culture-independent methods in different hospital settings globally.”

SC3: L117, replace "microbiome community" with "microbial community".

Response: Thank you for the suggestion. The sentence now reads:

Line 113-116: “Our main hypotheses were that (i) the bacterial microbial community could change before and after the patient occupancy and vary with hospital areas and (ii) several microbes can be associated with specific hospital areas.”

SC4: L172, why did you use the water control? Why didn't you include an unused swab also as a control?

Response: In our experience, the most significant contamination arises from the library preparation steps and carryover from prior sequencing runs. We try to minimize this by rotating our indices between sequencing runs. The water control allows us to detect these types of contamination. We agree that an unused swab control would have the added benefit of detecting potential contamination from the swab manufacturer. In our experience, the swabs and other sterile sample collection devices have not been a source of contamination in our laboratory. These devices are gamma irradiated by the manufacturer to minimize the amount of amplifiable bacterial DNA.

SC5: L183, sequence quality score above 10? To be honest, 10 is very low ... please indicate the average score of sequencing you obtained after quality filtering.

Response: We thank the reviewer for the insightful comment. Additional analyses and visualizations were performed to demonstrate the quality filter process. We would like to reassure the reviewer that the average read quality after filtering are 36.5 for the forward reads and 35.2 for the reverse reads, respectively, which indicates successful quality filtering.

Quality filtering and denoising was completed in a single step using the DADA2 pipeline in QIIME 2 which does not output the resulting quality scores. The denoising statistics provided from QIIME 2 were added for transparency. Additionally we simulated the same quality filtering and denoising using DADA2 in R so that the quality scores could

be examined (Table S1). The average read quality is 36.5 for the forward reads and 35.2 for the reverse reads which indicates successful quality filtering (Table S2, Figure S1).

FIG S1 (R3.SC5) Average read quality across all reads at each base during quality steps. The shaded region represents the 25th to 75th quartile range.

TABLE S1 Average Denoising Statistics per Sample (n=235).

Number of Unfiltered Reads		Percentage of Reads Remaining After Each Quality Step							
		Filtered		Denoised		Merged		Non-chimeric	
Average	SD	Average	SD	Average	SD	Average	SD	Average	SD
192,811	244,915	25.18%	14.27%	24.36%	13.92%	13.44%	10.35%	12.41%	8.78%

TABLE S2 Average Read Quality Through the Dada2 Pipeline (n = 235).

Pipeline Step	Number of Reads	Number of Bases	Quality Score	
			Average	SD
Forward Reads				
Unfiltered	42,645,418	300	30.2	4.61
Filtered	13,725,430	285	36.5	1.49
Denoised	47,093	285	36.5	1.63
Reverse Reads				
Unfiltered	42,645,418	300	27.6	5.71
Filtered	13,725,430	235	35.2	2.84
Denoised	30,227	235	35.0	3.03

SC6: *L191 suggests using greengene2 for taxonomic annotation, which was released recently and has a higher resolution at fine levels.*

Response: We thank the reviewer for the thoughtful suggestion. Additional taxonomic annotations were performed using Greengene2. We have referenced the Greengenes2 database in our taxonomic annotation. A comparison of the classification by SILVA and Greengenes2 for key ASVs discussed in the study is available in Table S10. Due to the increased potential for misclassification at the species level, we discuss phylogeny at the genera a family levels which are largely consistent between both methods of classification.

TABLE S10 Taxonomic Classification Comparison of Key ASVs.

ASV	Family			Genus			Species		
	SILVA	Greengenes2	SILVA	Greengenes2	SILVA	Greengenes2	SILVA	Greengenes2	
LASSO Influential ASVs									
Deinococcus ASV 35	Deinococcaceae	Deinococcaceae	Deinococcus	Deinococcus	Deinococcus	Deinococcus_B	bacterium_1227R	Deinococcus_B	
Meiothermus ASV 1	Thermaceae	Thermaceae_405955	Meiothermus	Meiothermus	Meiothermus	Meiothermus_B_405753	uncultured Meiothermus	Meiothermus	
Risungbineiella ASV 1	Thermactinomycetaceae	Thermactinomycetaceae	Risungbineiella	Risungbineiella	Risungbineiella	Risungbineiella	Risungbineiella	Risungbineiella	
Bdellovibrio ASV 1	Bdellovibrionaceae	Bdellovibrionaceae	Bdellovibrio	Bdellovibrio	Bdellovibrio	Bdellovibrio	Bdellovibrio	Bdellovibrio	
Rhizobium ASV 94	Rhizobiaceae	Rhizobiaceae_A_501059	Rhizobium	Rhizobium	Rhizobium	Mycoplana_499574	Mycoplana	Brevundimonas	
Brevundimonas ASV 17	Caulobacteraceae	Caulobacteraceae	Brevundimonas	Brevundimonas	Brevundimonas	Brevundimonas	Brevundimonas	Brevundimonas	
Novosphingobium ASV 12	Sphingomonadaceae	Sphingomonadaceae	Novosphingobium	Novosphingobium	Novosphingobium	Novosphingobium_485351	Novosphingobium	Novosphingobium	
Paracoccus ASV 5	Rhodobacteriaceae	Rhodobacteriaceae	Paracoccus	Paracoccus	Paracoccus	Paracoccus	Paracoccus	Paracoccus	
Oribacterium ASV 3	Lachnospiraceae	Lachnospiraceae	Oribacterium	Oribacterium	Oribacterium	Oribacterium	Oribacterium	Oribacterium	
Escherichia-Shigella ASV 1	Enterobacteriaceae	Enterobacteriaceae_A	Escherichia-Shigella	Escherichia-Shigella	Escherichia-Shigella	Escherichia-Shigella	Escherichia-Shigella	Escherichia-Shigella	
Escherichia-Shigella ASV 5	Enterobacteriaceae	Enterobacteriaceae_A	Escherichia-Shigella	Escherichia-Shigella	Escherichia-Shigella	Escherichia-Shigella	Escherichia-Shigella	Escherichia-Shigella	
Escherichia-Shigella ASV 2	Enterobacteriaceae	Enterobacteriaceae_A	Escherichia-Shigella	Escherichia-Shigella	Escherichia-Shigella	Escherichia-Shigella	Escherichia-Shigella	Escherichia-Shigella	
Brachyбактерium ASV 26	Dermabacteraceae	Dermabacteraceae	Brachyбактерium	Brachyбактерium	Brachyбактерium	Brachyбактерium	Brachyбактерium	Brachyбактерium	
Brachyбактерium ASV 2	Dermabacteraceae	Dermabacteraceae	Brachyбактерium	Brachyбактерium	Brachyбактерium	Brachyбактерium	Brachyбактерium	Brachyбактерium	
Kocuria ASV 22	Micrococcaceae	Micrococcaceae	Kocuria	Kocuria	Kocuria	Kocuria	Kocuria	Kocuria	
Kocuria ASV 27	Micrococcaceae	Micrococcaceae	Kocuria	Kocuria	Kocuria	Kocuria	Kocuria	Kocuria	
Fusobacterium ASV 6	Fusobacteriaceae	Fusobacteriaceae_993521	Fusobacterium	Fusobacterium	Fusobacterium	Fusobacterium_C	Fusobacterium	Fusobacterium	
Cutibacterium ASV 4	Protonibacteriaceae	Protonibacteriaceae	Cutibacterium	Cutibacterium	Cutibacterium	Cutibacterium	Cutibacterium	Cutibacterium	
LASSO Abundant ASVs									
Deinococcus ASV 3	Deinococcaceae	Deinococcaceae	Deinococcus	Deinococcus	Deinococcus	Deinococcus_B	Deinococcus	Deinococcus	
Blastocatella ASV 2	Blastocatellaceae	Blastocatellaceae_430966	Blastocatella	Blastocatella	Blastocatella	Alloprevotella	Alloprevotella	Alloprevotella	
Alloprevotella ASV 1	Prevotellaceae	Bacteroidaceae	Alloprevotella	Alloprevotella	Alloprevotella	Alloprevotella	Alloprevotella	Alloprevotella	
Bdellovibrio ASV 1	Bdellovibrionaceae	Bdellovibrionaceae	Bdellovibrio	Bdellovibrio	Bdellovibrio	Bdellovibrio	Bdellovibrio	Bdellovibrio	
Devosia ASV 5	Devosiaceae	Devosiaceae	Devosia	Devosia	Devosia	Devosia_A_502124	Devosia	Devosia	
Rhizobium ASV 2	Rhizobiaceae	Rhizobiaceae	Rhizobium	Rhizobium	Rhizobium	Rhizobium	Rhizobium	Rhizobium	
Microvinga ASV 1	Beijerinckiaceae	Beijerinckiaceae	Microvinga	Microvinga	Microvinga	Microvinga	Microvinga	Microvinga	
Phenyllobacterium ASV 1	Caulobacteraceae	Caulobacteraceae	Phenyllobacterium	Phenyllobacterium	Phenyllobacterium	Phenyllobacterium	Phenyllobacterium	Phenyllobacterium	
Sandaracinobacter ASV 1	Sphingomonadaceae	Sphingomonadaceae	Sandaracinobacter	Sandaracinobacter	Sandaracinobacter	Sandaracinobacter	Sandaracinobacter	Sandaracinobacter	
Sphingomonas ASV 86	Sphingomonadaceae	Sphingomonadaceae	Sphingomonas	Sphingomonas	Sphingomonas	Sphingomonas_L_486704	Sphingomonas	Sphingomonas	
Sphingobium ASV 9	Sphingomonadaceae	Sphingomonadaceae	Sphingobium	Sphingobium	Sphingobium	Sphingobium_A_485959	Sphingobium	Sphingobium	
Novosphingobium ASV 12	Sphingomonadaceae	Sphingomonadaceae	Novosphingobium	Novosphingobium	Novosphingobium	Novosphingobium_485351	Novosphingobium	Novosphingobium	
Amarcoccus ASV 2	Rhodobacteriaceae	Rhodobacteriaceae	Amarcoccus	Amarcoccus	Amarcoccus	Amarcoccus	Amarcoccus	Amarcoccus	
Paracoccus ASV 5	Rhodobacteriaceae	Rhodobacteriaceae	Paracoccus	Paracoccus	Paracoccus	Paracoccus	Paracoccus	Paracoccus	
Paracoccus ASV 85	Rhodobacteriaceae	Rhodobacteriaceae	Paracoccus	Paracoccus	Paracoccus	Paracoccus	Paracoccus	Paracoccus	
Rubellimicrobium ASV 5	Rhodobacteriaceae	Rhodobacteriaceae	Rubellimicrobium	Rubellimicrobium	Rubellimicrobium	Rubellimicrobium	Rubellimicrobium	Rubellimicrobium	
uncultured Rhodobacteraceae ASV 4	Rhodobacteriaceae	Rhodobacteriaceae	uncultured	uncultured	uncultured	uncultured	uncultured	uncultured	
Halomonas ASV 2	Halomonadaceae	Halomonadaceae_641030	Halomonas	Halomonas	Halomonas	Halomonas_C_640989	Halomonas	Halomonas	
Pseudomonas ASV 7	Pseudomonadaceae	Pseudomonadaceae	Pseudomonas	Pseudomonas	Pseudomonas	Pseudomonas_E_647464	Pseudomonas	Pseudomonas	
Moraxella ASV 3	Moraxellaceae	Moraxellaceae	Moraxella	Moraxella	Moraxella	Moraxella_C_651924	Moraxella	Moraxella	
Oribacterium ASV 3	Lachnospiraceae	Lachnospiraceae	Oribacterium	Oribacterium	Oribacterium	Oribacterium	Oribacterium	Oribacterium	
Johnsomella ASV 8	Lachnospiraceae	Lachnospiraceae	Johnsomella	Johnsomella	Johnsomella	Johnsomella	Johnsomella	Johnsomella	
Chryseobacterium ASV 46	Weeseellaceae	Weeseellaceae	Chryseobacterium	Chryseobacterium	Chryseobacterium	Chryseobacterium_796647	Chryseobacterium	Chryseobacterium	
Chryseobacterium ASV 20	Weeseellaceae	Weeseellaceae	Chryseobacterium	Chryseobacterium	Chryseobacterium	Chryseobacterium	Chryseobacterium	Chryseobacterium	
Elizabethkingia ASV 2	Weeseellaceae	Weeseellaceae	Elizabethkingia	Elizabethkingia	Elizabethkingia	Elizabethkingia	Elizabethkingia	Elizabethkingia	

ASV	Family			Genus			Species		
	SILVA	GreenGenes2	SILVA	GreenGenes2	SILVA	GreenGenes2	SILVA	GreenGenes2	
Elizabethkingia ASV 1	Weeksellaceae	Weeksellaceae	Elizabethkingia	Elizabethkingia	Elizabethkingia	Elizabethkingia	Elizabethkingia	Elizabethkingia	
Comamonas ASV 5	Comamonadaceae	Burkholderiaceae_A_592522	Comamonas	Comamonas	Comamonas_F_589250	Comamonas	Comamonas	Comamonas	
Azospira ASV 2	Rhodocyclaceae	Rhodocyclaceae	Azospira	Azospira	Azospira_A	Azospira	Azospira	Azospira	
Escherichia-Shigella ASV 1	Enterobacteriaceae	Enterobacteriaceae_A	Escherichia-Shigella	Escherichia-Shigella	Escherichia-Shigella	Escherichia-Shigella	Escherichia-Shigella	Escherichia-Shigella	
Escherichia-Shigella ASV 5	Enterobacteriaceae	Enterobacteriaceae_A	Escherichia-Shigella	Escherichia-Shigella	Escherichia-Shigella	Escherichia-Shigella	Escherichia-Shigella	Escherichia-Shigella	
Escherichia-Shigella ASV 2	Enterobacteriaceae	Enterobacteriaceae_A	Escherichia-Shigella	Escherichia-Shigella	Escherichia-Shigella	Escherichia-Shigella	Escherichia-Shigella	Escherichia-Shigella	
Pseudomonas ASV 199	Pseudomonadaceae	Pseudomonadaceae	Pseudomonas	Pseudomonas	Pseudomonas_A	Pseudomonas	Pseudomonas	Pseudomonas	
Pseudomonas ASV 138	Pseudomonadaceae	Pseudomonadaceae	Pseudomonas	Pseudomonas	Pseudomonas_E_648040	Pseudomonas	Pseudomonas	Pseudomonas	
Pseudomonas ASV 30	Pseudomonadaceae	Pseudomonadaceae	Pseudomonas	Pseudomonas	Pseudomonas_E_647464	Pseudomonas	Pseudomonas	Pseudomonas	
Pseudomonas ASV 105	Pseudomonadaceae	Pseudomonadaceae	Pseudomonas	Pseudomonas	Pseudomonas_A	Pseudomonas	Pseudomonas	Pseudomonas	
Pseudomonas ASV 142	Pseudomonadaceae	Pseudomonadaceae	Pseudomonas	Pseudomonas	Pseudomonas_B_650451	Pseudomonas	Pseudomonas	Pseudomonas	
uncultured Acetobacteraceae ASV 1	Acetobacteraceae	Acetobacteraceae	uncultured	uncultured	uncultured	uncultured	uncultured	uncultured	
Paratruncococcus ASV 67	Acetobacteraceae	Acetobacteraceae	Paratruncococcus	Paratruncococcus	Paratruncococcus_506950	Paratruncococcus	Paratruncococcus	Paratruncococcus	
Brachyobacterium ASV 2	Dermabacteriaceae	Dermabacteriaceae	Brachyobacterium	Brachyobacterium	Brachyobacterium	Brachyobacterium	Brachyobacterium	Brachyobacterium	
Pseudodoclavibacter ASV 1	Microbacteriaceae	Microbacteriaceae	Pseudodoclavibacter	Pseudodoclavibacter	Pseudodoclavibacter_A_383705	Pseudodoclavibacter	Pseudodoclavibacter	Pseudodoclavibacter	
Frigoribacterium ASV 1	Microbacteriaceae	Microbacteriaceae	Frigoribacterium	Frigoribacterium	Frigoribacterium	Frigoribacterium	Frigoribacterium	Frigoribacterium	
Micrococcus ASV 8	Micrococcaceae	Micrococcaceae	Micrococcus	Micrococcus	Micrococcus	Micrococcus	Micrococcus	Micrococcus	
Paenarthrobacter ASV 1	Micrococcaceae	Micrococcaceae	Paenarthrobacter	Paenarthrobacter	Paenarthrobacter	Paenarthrobacter	Paenarthrobacter	Paenarthrobacter	
Brevibacterium ASV 1	Brevibacteriaceae	Brevibacteriaceae	Brevibacterium	Brevibacterium	Brevibacterium	Brevibacterium	Brevibacterium	Brevibacterium	
Kocuria ASV 22	Micrococcaceae	Micrococcaceae	Kocuria	Kocuria	Kocuria	Kocuria	Kocuria	Kocuria	
Kocuria ASV 15	Micrococcaceae	Micrococcaceae	Kocuria	Kocuria	Kocuria	Kocuria	Kocuria	Kocuria	
Brevibacterium ASV 3	Brevibacteriaceae	Brevibacteriaceae	Brevibacterium	Brevibacterium	Brevibacterium	Brevibacterium	Brevibacterium	Brevibacterium	
Fusobacterium ASV 6	Fusobacteriaceae	Fusobacteriaceae_993521	Fusobacterium	Fusobacterium	Fusobacterium_C	Fusobacterium	Fusobacterium	Fusobacterium	
Leptotrichia ASV 197	Leptotrichiaceae	Leptotrichiaceae	Leptotrichia	Leptotrichia	Leptotrichia	Leptotrichia	Leptotrichia	Leptotrichia	
Cutibacterium ASV 4	Propionibacteriaceae	Propionibacteriaceae	Cutibacterium	Cutibacterium	Cutibacterium	Cutibacterium	Cutibacterium	Cutibacterium	
Corynebacterium ASV 8	Corynebacteriaceae	Corynebacteriaceae	Corynebacterium	Corynebacterium	Corynebacterium	Corynebacterium	Corynebacterium	Corynebacterium	
Corynebacterium ASV 683	Corynebacteriaceae	Corynebacteriaceae	Corynebacterium	Corynebacterium	Corynebacterium	Corynebacterium	Corynebacterium	Corynebacterium	
Corynebacterium ASV 448	Corynebacteriaceae	Corynebacteriaceae	Corynebacterium	Corynebacterium	Corynebacterium	Corynebacterium	Corynebacterium	Corynebacterium	
Corynebacterium ASV 167	Corynebacteriaceae	Corynebacteriaceae	Corynebacterium	Corynebacterium	Corynebacterium	Corynebacterium	Corynebacterium	Corynebacterium	
uncultured Corynebacteriaceae ASV 2	Corynebacteriaceae	Corynebacteriaceae	uncultured	uncultured	uncultured	uncultured	uncultured	uncultured	
Corynebacterium ASV 155	Corynebacteriaceae	Corynebacteriaceae	Corynebacterium	Corynebacterium	Corynebacterium	Corynebacterium	Corynebacterium	Corynebacterium	
Corynebacterium ASV 30	Corynebacteriaceae	Corynebacteriaceae	Corynebacterium	Corynebacterium	Corynebacterium	Corynebacterium	Corynebacterium	Corynebacterium	
Corynebacterium ASV 101	Corynebacteriaceae	Corynebacteriaceae	Corynebacterium	Corynebacterium	Corynebacterium	Corynebacterium	Corynebacterium	Corynebacterium	
Core ASVs									
Rhizobium ASV 1	Rhizobiaceae	Rhizobiaceae	Rhizobium	Rhizobium	Rhizobium	Rhizobium	Rhizobium	Rhizobium	
Brevundimonas ASV 3	Caulobacteraceae	Caulobacteraceae	Brevundimonas	Brevundimonas	Brevundimonas	Brevundimonas	Brevundimonas	Brevundimonas	
Paracoccus ASV 2	Rhodobacteriaceae	Rhodobacteriaceae	Paracoccus	Paracoccus	Paracoccus	Paracoccus	Paracoccus	Paracoccus	
Pseudomonas ASV 8	Pseudomonadaceae	Pseudomonadaceae	Pseudomonas	Pseudomonas	Pseudomonas_E_647464	Pseudomonas	Pseudomonas	Pseudomonas	
Enhydrobacter ASV 1	Moraxellaceae	Moraxellaceae	Enhydrobacter	Enhydrobacter	Enhydrobacter	Enhydrobacter	Enhydrobacter	Enhydrobacter	
Anaerococcus ASV 2	Peptostreptococcales-Tissierell	Peptostreptococcales-Tissierell	Anaerococcus	Anaerococcus	Anaerococcus	Anaerococcus	Anaerococcus	Anaerococcus	
Peptoniphilus ASV 1	Peptoniphilaceae	Peptoniphilaceae	Peptoniphilus	Peptoniphilus	Peptoniphilus_A	Peptoniphilus	Peptoniphilus	Peptoniphilus	
Chryseobacterium ASV 1	Weeksellaceae	Weeksellaceae	Chryseobacterium	Chryseobacterium_796647	Chryseobacterium	Chryseobacterium	Chryseobacterium	Chryseobacterium	
Chryseobacterium ASV 2	Weeksellaceae	Weeksellaceae	Chryseobacterium	Chryseobacterium_796647	Chryseobacterium	Chryseobacterium	Chryseobacterium	Chryseobacterium	
Empedobacter ASV 1	Weeksellaceae	Weeksellaceae	Empedobacter	Empedobacter_790298	Empedobacter	Empedobacter	Empedobacter	Empedobacter	
Comamonas ASV 1	Burkholderiaceae_A_592522	Comamonadaceae	Comamonas	Comamonas	Comamonas_F_589250	Comamonas	Comamonas	Comamonas	
Pseudomonas ASV 5	Pseudomonadaceae	Pseudomonadaceae	Pseudomonas	Pseudomonas	Pseudomonas_E_648040	Pseudomonas	Pseudomonas	Pseudomonas	
Pseudomonas ASV 4	Pseudomonadaceae	Pseudomonadaceae	Pseudomonas	Pseudomonas	Pseudomonas_E_648040	Pseudomonas	Pseudomonas	Pseudomonas	
Pseudomonas ASV 2	Pseudomonadaceae	Pseudomonadaceae	Pseudomonas	Pseudomonas	Pseudomonas	Pseudomonas	Pseudomonas	Pseudomonas	
Roseomonas ASV 1	Acetobacteraceae	Acetobacteraceae	Roseomonas	Roseomonas	Roseomonas_A_507160	Roseomonas	Roseomonas	Roseomonas	

ASV	Family			Genus			Species		
	SILVA	Greengenes2	SILVA	Greengenes2	SILVA	Greengenes2	SILVA	Greengenes2	
Cutibacterium ASV 3	Propionibacteriaceae	Propionibacteriaceae	Cutibacterium	Cutibacterium	Cutibacterium	Cutibacterium	Cutibacterium	Cutibacterium	
Micrococcus ASV 1	Micrococcaceae	Micrococcaceae	Micrococcus	Micrococcus	Micrococcus	Micrococcus	Micrococcus	Micrococcus	
Micrococcus ASV 3	Micrococcaceae	Micrococcaceae	Micrococcus	Micrococcus	Micrococcus	Micrococcus	Micrococcus	Micrococcus	
Corynebacterium ASV 1	Mycobacteriaceae	Corynebacteriaceae	Corynebacterium	Corynebacterium	Corynebacterium	Corynebacterium	Corynebacterium	Corynebacterium	
Rothia ASV 1	Micrococcaceae	Micrococcaceae	Rothia	Rothia	Rothia	Rothia	Rothia	Rothia	
Rothia ASV 2	Micrococcaceae	Micrococcaceae	Rothia	Rothia	Rothia	Rothia	Rothia	Rothia	
Fusobacterium ASV 1	Fusobacteriaceae_993521	Fusobacteriaceae	Fusobacterium_C	Fusobacterium_C	Fusobacterium_C	Fusobacterium	Fusobacterium_C	Fusobacterium	
Cutibacterium ASV 1	Propionibacteriaceae	Propionibacteriaceae	Cutibacterium	Cutibacterium	Cutibacterium	Cutibacterium	Cutibacterium	Cutibacterium	
Cutibacterium ASV 2	Propionibacteriaceae	Propionibacteriaceae	Cutibacterium	Cutibacterium	Cutibacterium	Cutibacterium	Cutibacterium	Cutibacterium	
Lawsonella ASV 1	Mycobacteriaceae	Corynebacteriaceae	Lawsonella	Lawsonella	Lawsonella	Lawsonella	Lawsonella	Lawsonella	

SC7: L198 sampling date is also one of the metadata.

Response: Thank you for bringing up this oversight. The sentence now reads:

Line 196-197: “The metadata included locations (towers, levels, and room numbers), area types (six areas as described earlier), room types, surfaces, and sampling date.”

SC8: L219, why didn't you apply weighted and unweighted unifracs distance metrics, which were widely used in the beta diversity of 16S amplicon data analysis?

Response: We have added PCoA plots based on weighted and unweighted UniFrac and Jaccard distances in the Supplementary (Figures S6). These diversity metrics lead us to the same conclusion that there are two distinct clusters: the high diversity group, and the low diversity group.

FIG S1 (R3.SC8) PCoA plots of all door handle, keyboard, and office electronic samples taken after the hospital opened for inpatient care (n = 120) and color coded by the area type of where the sample was taken including multivariate t distribution ellipses of the low and high diversity groups. Sample sizes: Pathology lab: n = 20, IPAC offices: n = 42, Microbiology lab: n = 8, PICU: n = 19, NICU: n = 15, Pediatric surgery ward: n = 16. PERMANOVA test: Ho: The centroids of the groups are the same. Ha: The centroids of the groups are not the same. p = 0.001. Ho can be rejected. Solid symbols are area types in the low diversity group, and open symbols are in the high diversity group. A) Unweighted Unifrac distance, B) Weighted Unifrac distance, C) Jaccard distance.

SC9: *L219-228, in the statistics of permanova, because you have multiple factors, how did you control the influence of others when you were testing the significance of one factor? Please illustrate clearly in the method section. It is good that you converted the categorical factor "sampling data" to a numeric factor "number of days after opening" in statistics.*

Response: The PERMANOVA test included all factors at once in a single test, therefore no additional corrections were needed. The reviewer's comment helped us see the need to more clearly describe how the PERMANOVA test was set up. We've now specified:

Line 224-226: "In the analysis of door handle, keyboard, and office electronic samples taken after opening, the groups were defined by area type, surface category, and number of days after opening **in a single statistical test.**"

SC10: *L246-261 suggests moving to the method section.*

Response: We thank the reviewers for this suggestion. Because we used a novel approach of engaging healthcare workers for sampling, the sampling coverage is a notable result of our study. In order to emphasize this point, we've added emphasis to it in the discussion:

Line 430-434: "**Healthcare workers chose which high-touch surfaces to sample which increased their awareness of potential microbial reservoirs in their workspace, evidenced by the diverse types of surfaces (Table S3). Potential issues of sampling bias were mitigated by repeated sampling over time and analyzing the most frequently sampled surfaces.**"

SC11: *L268, is there a reference for the core microbiome at a prevalence cutoff of 40%? In addition to the prevalence, the relative abundance is also important for defining the*

core microbiome; some previous studies have suggested using 1% as the cutoff for the core taxa, if I remember correctly.

Response: In our literature search, we have found that it is very common practice for the authors to specify a prevalence cutoff in their dataset. By examining the prevalence histogram of our own dataset (Figure S3), we set a 50% cutoff (a correction in the manuscript) which allowed us to identify core microbes without being too conservative given the sampling size.

Reference: Neu, A. T.; Allen, E. E.; Roy, K. Defining and Quantifying the Core Microbiome: Challenges and Prospects. *Proceedings of the National Academy of Sciences* **2021**, *118* (51), e2104429118. <https://doi.org/10.1073/pnas.2104429118>.

FIG S3 (R3.SC8) Histogram of the log-transformed prevalence of ASVs in all samples. The core population was defined as the ASVs present in at least 50% of the samples which yielded 25 ASVs which constitute 53.7% of the total relative abundance.

SC12: *L270-291, among such many areas, which area and which sample types contain the highest relative abundance of pathogens? It suggests using a heat map showing the*

results.

Response: We agree with the reviewer that the area associations with opportunistic pathogens is an important and useful topic. We have the data from cultures and the corresponding 16S genera and show it in a heatmap (Figure 5A).

FIG 5 (R3.SC12) A) Heatmap of the relative CFU of each species cultured from all surface samples taken in the IPAC offices, PICU, pathology lab, microbiology lab, and pediatric surgery ward after the hospital opened (n = 160). Hierarchical clustering was performed with all samples using the Ward

method. The annotation bar indicates the area type of the sample. B) The prevalences and C) relative abundances of each genera in each sample as measured by 16S rRNA gene sequencing and as measured by culture analysis. The weak correlation demonstrates both the compatibility of the methods and added value of 16S sequencing for wider data capture.

SC13: *Table 1 and table 2 suggest to be put as supplementary files.*

Response: Table 1 and Table 2 have been revised to be more concise, and the pairwise statistical analyses have been moved to the Supplementary (Table S6 and S7)

TABLE 1 Comparisons of means in alpha diversity metrics.

Grouping	Observed Richness		Shannon Index		Pielou's Evenness	
	p-value	p-adjusted	p-value	p-adjusted	p-value	p-adjusted
A) Wilcoxon rank sum tests on means of alpha diversity metrics in NICU patient rooms before and after opening.						
Before opening (n = 14) vs. After opening (n = 14)	1.77×10^{-1}	-	3.26×10^{-1}	-	5.42×10^{-1}	-
B) Kruskal-Wallis tests on means of alpha diversity metrics in different surface categories and area types in all door handle, keyboard, and office electronic samples taken after the hospital opened for inpatient care (n = 120). ¹						
Surface Category	1.10×10^{-1}	1.10×10^{-1}	2.86×10^{-1}	2.86×10^{-1}	5.74×10^{-1}	5.74×10^{-1}
Area Type	9.84×10^{-15}	1.97×10^{-14}	8.57×10^{-14}	1.71×10^{-13}	1.67×10^{-10}	3.35×10^{-10}

307

¹ Bonferroni correction was used to adjust p-values, and significant p-values are bolded ($\alpha = 0.05$).

TABLE S6 Pairwise Mann-Whitney U tests on means of alpha diversity metrics on the various area types in all door handle, keyboard, and office electronic samples taken after the hospital opened for inpatient care (n = 120).¹

Pairing	Observed Richness		Shannon Index		Pielou's Evenness	
	p-value	p-adjusted	p-value	p-adjusted	p-value	p-adjusted
Area Type						
Microbiology Lab vs IPAC Offices	1.05×10^{-5}	5.26×10^{-5}	1.49×10^{-8}	7.45×10^{-8}	6.74×10^{-5}	2.02×10^{-4}
Microbiology Lab vs NICU	1.30×10^{-2}	1.77×10^{-2}	5.55×10^{-4}	1.04×10^{-3}	2.73×10^{-4}	6.83×10^{-4}
Microbiology Lab vs Pathology Lab	0.780	0.835	0.901	0.901	0.601	0.693
Microbiology Lab vs Pediatric Surgery Ward	9.44×10^{-4}	1.77×10^{-3}	1.66×10^{-3}	2.76×10^{-3}	1.59×10^{-2}	2.66×10^{-2}
Microbiology Lab vs PICU	0.652	0.752	0.389	0.449	0.938	0.950
NICU vs IPAC Offices	4.92×10^{-5}	1.47×10^{-4}	0.253	0.316	0.950	0.950
NICU vs Pediatric Surgery Ward	9.08×10^{-3}	1.36×10^{-2}	0.682	0.731	0.233	0.317
Pathology Lab vs IPAC Offices	2.66×10^{-10}	3.99×10^{-9}	2.11×10^{-14}	3.16×10^{-13}	3.39×10^{-9}	5.08×10^{-8}
Pathology Lab vs NICU	7.15×10^{-4}	1.53×10^{-3}	5.53×10^{-6}	1.80×10^{-5}	2.73×10^{-6}	1.36×10^{-5}
Pathology Lab vs Pediatric Surgery Ward	3.24×10^{-5}	1.22×10^{-4}	6.01×10^{-6}	1.80×10^{-5}	4.59×10^{-4}	9.83×10^{-4}
Pathology Lab vs PICU	0.173	0.216	0.120	0.180	0.309	0.386
Pediatric Surgery Ward vs IPAC Offices	0.965	0.965	0.174	0.237	6.28×10^{-2}	9.42×10^{-2}
PICU vs IPAC Offices	2.44×10^{-9}	1.83×10^{-8}	6.23×10^{-12}	4.67×10^{-11}	6.22×10^{-8}	4.67×10^{-7}
PICU vs NICU	7.96×10^{-3}	1.33×10^{-2}	5.19×10^{-5}	1.30×10^{-4}	2.31×10^{-5}	8.68×10^{-5}
PICU vs Pediatric Surgery Ward	8.69×10^{-5}	2.17×10^{-4}	7.43×10^{-5}	1.59×10^{-4}	3.49×10^{-3}	6.54×10^{-3}

¹ Bonferroni correction was used to adjust p-values, and significant p-values are bolded ($\alpha = 0.05$).

TABLE 2 Comparisons of centroids in principal coordinate analysis (PCoA).¹

Grouping	Df	PERMANOVA			PERMDISP		
		F value	R ²	p-value	F value	p-value	p-adjusted
A) PERMANOVA test on centroids and PERMDISP test on dispersions of PCoA in NICU patient rooms before and after opening.							
Hospital Opening	1	7.23 × 10 ⁻¹	2.70 × 10 ⁻²	7.32 × 10 ⁻¹	9.48 × 10 ⁻²	7.69 × 10 ⁻¹	-
B) PERMANOVA tests on centroids and PERMDISP test on dispersions of PCoA in different surface categories, area types, and days after opening in all door handle, keyboard, and office electronic samples taken after the hospital opened for inpatient care (n = 120). ²							
Surface Category	2	2.42	2.89 × 10 ⁻²	6.00 × 10⁻³	4.53 × 10 ⁻²	9.45 × 10 ⁻¹	0.945
Area Type	5	1.01 × 10 ¹	3.01 × 10 ⁻¹	1.00 × 10⁻³	1.50	2.07 × 10 ⁻¹	0.594
Days After Opening	1	8.95 × 10 ⁻¹	5.36 × 10 ⁻³	4.40 × 10 ⁻¹	1.05	3.96 × 10 ⁻¹	0.594

¹ASV relative abundances were square-root transformed prior to calculating the dissimilarity.

308

²Bonferroni correction was used to adjust p-values, and significant p-values are bolded ($\alpha = 0.05$).

Table S7. Pairwise PERMANOVA tests on centroids of PCoA on the various area types in different surface categories, area types, and days after opening in all door handle, keyboard, and office electronic samples taken after the hospital opened for inpatient care (n = 120).^{1,2}

Grouping	Df	PERMANOVA			
		F value	R ²	p-value	p-adjusted
Surface Category					
Keyboard vs Door Handle	1	1.88	2.14 × 10 ⁻²	5.80 × 10 ⁻²	8.70 × 10 ⁻²
Keyboard vs Office Electronics	1	1.16	1.86 × 10 ⁻²	0.231	0.231
Door Handle vs Office Electronics	1	2.04	2.29 × 10 ⁻²	5.40 × 10 ⁻²	8.70 × 10 ⁻²
Area Type					
NICU vs IPAC Offices	1	3.13	5.38 × 10 ⁻²	1.00 × 10 ⁻³	1.36 × 10 ⁻³
NICU vs Pathology Lab	1	14.7	0.309	1.00 × 10 ⁻³	1.36 × 10 ⁻³
NICU vs Microbiology Lab	1	16.0	0.433	1.00 × 10 ⁻³	1.36 × 10 ⁻³
NICU vs PICU	1	18.5	0.366	1.00 × 10 ⁻³	1.36 × 10 ⁻³
NICU vs Pediatric Surgery Ward	1	2.62	8.28 × 10 ⁻²	1.00 × 10 ⁻³	1.36 × 10 ⁻³
IPAC Offices vs Pathology Lab	1	20.4	0.253	1.00 × 10 ⁻³	1.36 × 10 ⁻³
IPAC Offices vs Microbiology Lab	1	16.2	0.253	1.00 × 10 ⁻³	1.36 × 10 ⁻³
IPAC Offices vs PICU	1	23.4	0.284	1.00 × 10 ⁻³	1.36 × 10 ⁻³
IPAC Offices vs Pediatric Surgery Ward	1	1.73	2.99 × 10 ⁻²	2.10 × 10 ⁻²	2.25 × 10 ⁻²
Pathology Lab vs Microbiology Lab	1	1.95	6.99 × 10 ⁻²	1.70 × 10 ⁻²	1.96 × 10 ⁻²
Pathology Lab vs PICU	1	2.11	5.38 × 10 ⁻²	2.00 × 10 ⁻³	2.50 × 10 ⁻³
Pathology Lab vs Pediatric Surgery Ward	1	8.62	0.202	1.00 × 10 ⁻³	1.36 × 10 ⁻³
Microbiology Lab vs PICU	1	1.27	4.84 × 10 ⁻²	0.1	0.1
Microbiology Lab vs Pediatric Surgery Ward	1	8.06	0.268	1.00 × 10 ⁻³	1.36 × 10 ⁻³
PICU vs Pediatric Surgery Ward	1	9.80	0.229	1.00 × 10 ⁻³	1.36 × 10 ⁻³

¹ASV relative abundances were square-root transformed prior to calculating the dissimilarity.

²Bonferroni correction was used to adjust p-values, and significant p-values are bolded ($\alpha = 0.05$).

Discussion

SC14: *L448, too absolute, the healthcare workers can bring in external microbes in the NICU.*

Response: In this part of the discussion, we are focusing on the patients as external/transient occupants whereas healthcare workers are assumed to be consistently in the environment and relatively stable. In order to clarify, the sentence was added,

Line 445-448: “If we consider the healthcare workers as consistent occupants of the hospital who practice stringent infection control measures, we can assume variability between areas to be caused by differences in patient occupancy.”

Re: Spectrum00296-24R1 (Intra-hospital Microbiome Variability Is Driven by Accessibility and Clinical Activities)

Dear Dr. Fangqiong Ling:

Thanks for carefully addressing the Reviewers' points. I would like to congratulate all authors on the acceptance of their paper for publication in Spectrum!

Your manuscript has been accepted, and I am forwarding it to the ASM production staff for publication. Your paper will first be checked to make sure all elements meet the technical requirements. ASM staff will contact you if anything needs to be revised before copyediting and production can begin. Otherwise, you will be notified when your proofs are ready to be viewed.

Sincerely,
Jan Claesen
Editor
Microbiology Spectrum